# Bacterial diet affects the age-dependent decline of associative learning in *Caenorhabditis elegans*

**Satoshi Higurashi[1,2†], Sachio Tsukada[1,2†], Binta Maria Aleogho[2,3,4†], Joo Hyun Park[2], Yana Al-Hebri[2], Masaru Tanaka[1,2], Shunji Nakano[3], Ikue Mori[3], Kentaro Noma[2,3,4*]**

[1]Milk Science Research Institute, Megmilk Snow Brand Co. Ltd., Saitama, Japan; [2]Group of Nutritional Neuroscience, Neuroscience Institute, Graduate School of Science, Nagoya University, Nagoya, Japan; [3]Group of Molecular Neurobiology, Neuroscience Institute, Graduate School of Science, Nagoya University, Nagoya, Japan; [4]Group of Microbial Motility, Department of Biological Science, Division of Natural Science, Graduate school of Science, Nagoya University, Nagoya, Japan

**Abstract** The causality and mechanism of dietary effects on brain aging are still unclear due to the long time scales of aging. The nematode *Caenorhabditis elegans* has contributed to aging research because of its short lifespan and easy genetic manipulation. When fed the standard laboratory diet, *Escherichia coli*, *C. elegans* experiences an age-dependent decline in temperature–food associative learning, called thermotaxis. To address if diet affects this decline, we screened 35 lactic acid bacteria as alternative diet and found that animals maintained high thermotaxis ability when fed a clade of *Lactobacilli* enriched with heterofermentative bacteria. Among them, *Lactobacillus reuteri* maintained the thermotaxis of aged animals without affecting their lifespan and motility. The effect of *Lb. reuteri* depends on the DAF-16 transcription factor functioning in neurons. Furthermore, RNA sequencing analysis revealed that differentially expressed genes between aged animals fed different bacteria were enriched with DAF-16 targets. Our results demonstrate that diet can impact brain aging in a *daf-16*-dependent manner without changing the lifespan.

## Editor's evaluation

This important work focuses on the impact of diet on age-dependent behavior decline, showing that worms grown on *E. coli*, a common laboratory diet, lose their thermotaxis ability as they grow older, and that a diet of LAB partially rescue this effect. The evidence supporting the claims is solid, although the mechanism for the effects is not yet fully characterized. The work will be of interest to scientists interested in aging, behavior, diet, and potentially the microbiome.

## Introduction

Human life expectancy has increased since the nineteenth century (*Dong et al., 2016*), which has led to the social problem related to age-dependent cognitive dysfunction. Although human studies suggest that genetic background, diet, and lifestyle might affect brain aging, the causality and mechanism of how they affect brain aging remain unclear (*Deary et al., 2009*).

The nematode *Caenorhabditis elegans* is ideal for addressing the mechanism of age-related phenotypes because of the 2- to 3-week lifespan and the variety of available genetic tools. In *C. elegans*, the age-related phenotypes can be readily separable from the organismal lifespan by directly measuring the lifespan. In the past decades, studies using *C. elegans* have contributed to aging research by

**\*For correspondence:**
noma.kentaro.f1@f.mail.nagoya-u.ac.jp

[†]These authors contributed equally to this work

revealing the molecular mechanism of how dietary restriction, insulin-like signaling, and germline stem cells affect organismal lifespan (*Wolff and Dillin, 2006*; *Mack et al., 2018*). Like mammals, *C. elegans* experiences age-dependent functional changes in the nervous system (*Stein and Murphy, 2012*). Aged animals are defective in locomotion (*Mulcahy et al., 2013*; *Hahm et al., 2015*), mechanosensory response (*Beck and Rankin, 1993*), chemotaxis (*Leinwand et al., 2015*), thermotaxis (*Murakami et al., 2005*; *Murakami and Murakami, 2005*; *Huang et al., 2020*), and food–butanone associative learning (*Kauffman et al., 2010*). Age-dependent memory decline in the food–butanone association is ameliorated in the mutant of *nkat-1* encoding kynurenic acid-synthesizing enzyme (*Vohra et al., 2018*). Overactivation of Gα signaling in AWC sensory neurons also maintains the ability to form memory in aged animals in the food–butanone association (*Arey et al., 2018*). These emerging evidence suggest that genetic manipulations can prevent age-dependent functional decline in the nervous system.

Compared to genetic manipulations, the modification of diet can be easily applicable to our daily lives. Studies in humans and mice imply that diet affects the cognitive decline in aged animals (*Joseph et al., 2009*; *Vauzour et al., 2017*). Here, we use *C. elegans* to address the dietary effect on the age-dependent behavioral decline and its underlying mechanism. In laboratories, *C. elegans* is maintained monoxenically with a uracil auxotroph *Escherichia coli* strain, OP50, as the standard diet (*Brenner, 1974*). On the other hand, *C. elegans* in natural habitat eats a wide variety of bacteria (*Berg et al., 2016*; *Dirksen et al., 2016*; *Samuel et al., 2016*; *Zhang et al., 2017*; *Johnke et al., 2020*). These bacteria affect the physiology of *C. elegans*, such as growth rate, reproduction, and sensory behavior (*Dirksen et al., 2016*; *Samuel et al., 2016*; *O'Donnell et al., 2020*). However, the effect of different bacteria on the behavioral decline during aging is unexplored. Among the potential bacterial diet for *C. elegans* in natural habitat (*Berg et al., 2016*; *Dirksen et al., 2016*; *Samuel et al., 2016*), we focused on lactic acid bacteria (LAB), which are the most common probiotics for humans (*Hill et al., 2014*). LAB, such as *Lactobacilli* (*Lb.*) and *Bifidobacteria* (*B.*), are Gram-positive, non-spore-forming bacteria that produce lactic acid from carbohydrates as the primary metabolic product. Depending on the species, LAB have various effects on *C. elegans* physiology. *Lb. gasseri*, *B. longum*, and *B. infantis* extend lifespan in *C. elegans* (*Komura et al., 2013*; *Nakagawa et al., 2016*; *Zhao et al., 2017*). On the other hand, *Lb. helveticus* does not increase the lifespan (*Nakagawa et al., 2016*). Even in the same species, different strains have different effects on lifespan, body size, and locomotion (*Wang et al., 2020*). In *C. elegans*, LAB modulate evolutionarily conserved genetic pathways such as the insulin/insulin-like growth factor-1 (IGF-1) signaling (IIS) pathway (*Grompone et al., 2012*; *Sugawara and Sakamoto, 2018*), which consists of the insulin receptor DAF-2, phosphoinositide 3 (PI3) kinase cascade, and the downstream transcription factor DAF-16 (*Kenyon et al., 1993*; *Lin et al., 1997*). DAF-16 is the sole *C. elegans* ortholog of mammalian FOXO transcription factor and is involved in various biological processes (*Stein and Murphy, 2012*; *Tissenbaum, 2018*). Moreover, *daf-16* is involved in the age-dependent modulation of isothermal tracking behavior in *C. elegans* with regular *E. coli* diet (*Murakami et al., 2005*).

To comprehensively understand the effect of LAB, we screened 35 different LAB species, including some subspecies. We examined the age-dependent functional decline of thermotaxis behavior, which reflects associative learning between temperature and food (*Hedgecock and Russell, 1975*; *Mori and Ohshima, 1995*). We demonstrate that *C. elegans* fed *Lactobacilli* in a clade maintained the thermotaxis behavior when aged, while *E. coli*-fed animals did not. Among those *Lactobacilli*, *Lb. reuteri* maintained the thermotaxis ability of aged animals without affecting the organismal lifespan or locomotion. The effect of *Lb. reuteri* on the thermotaxis of aged animals depends on the DAF-16 transcription factor functioning in neurons.

## Results

### *C. elegans* thermotaxis behavior declines with age

After being cultivated with food at a temperature within the physiological range (15–25°C), *C. elegans* migrates toward and stays at the past cultivation temperature ($T_{cult}$) on a linear thermal gradient without food (*Figure 1A*). This behavior is called thermotaxis (*Hedgecock and Russell, 1975*; *Mori and Ohshima, 1995*). To see the effect of aging on thermotaxis, we cultivated animals at 20°C with an *E. coli* strain, OP50 (hereafter, *E. coli*, unless otherwise noted), commonly used in laboratory

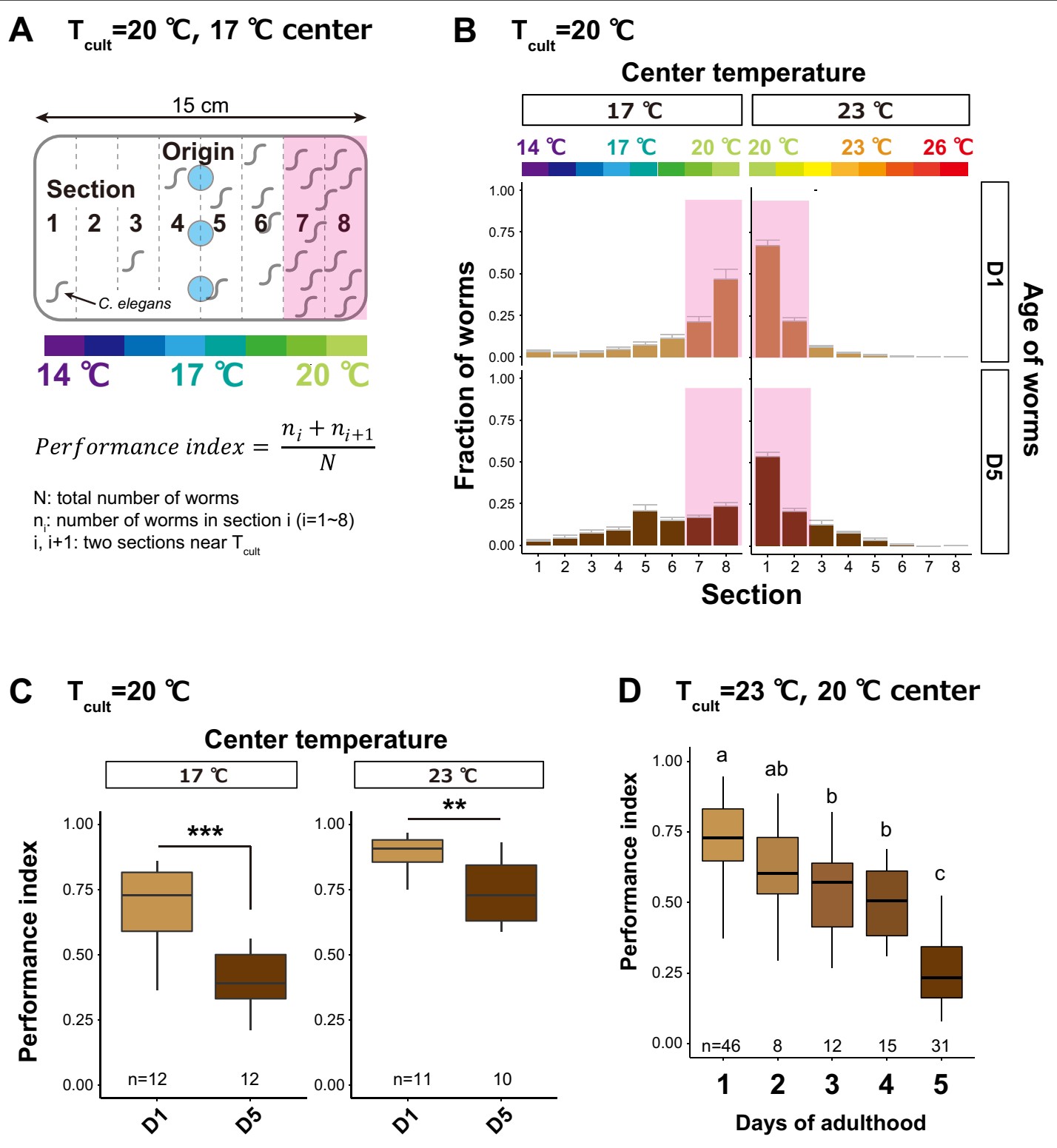

**Figure 1.** Thermotaxis performance declines with age. (**A**) Schematic of thermotaxis assay. Animals were placed at light blue circles on a thermal gradient without food. The pink rectangle indicates the sections around the $T_{cult}$. After 1 hr, the number of animals in each section was counted to calculate the thermotaxis performance index using the indicated formula. (**B, C**) Age-dependent changes in thermotaxis behavior. D1 and D5 animals were cultivated with *E. coli* at 20°C and placed at the center of a 14–20 or 20–26°C gradient. (**B**) Distributions of animals (pink rectangle: the sections around the $T_{cult}$) on the thermotaxis plates. (**C**) Box plots of thermotaxis performance indices. The number of experiments is shown. Statistics: Student's *t*-test compared to D1 adults. **p < 0.01, ***p < 0.001. (**D**) Box plots summarizing thermotaxis performance indices of animals at different ages. Animals

*Figure 1 continued on next page*

*Figure 1 continued*

were cultivated with *E. coli* at 23°C and placed at the center of a 17–23°C gradient. The number of experiments is shown. Statistics: The mean indices marked with distinct alphabets are significantly different (p < 0.05) according to one-way analysis of variance (ANOVA) followed by Tukey–Kramer test.

The online version of this article includes the following source data and figure supplement(s) for figure 1:

**Source data 1.** Thermotaxis assays with 17 or 23°C center.

**Source data 2.** Thermotaxis assays at different ages.

**Figure supplement 1.** Survival curve of animals cultivated at different temperatures.

**Figure supplement 1—source data 1.** Lifespan at different temperatures.

**Figure supplement 2.** Distributions of animals at different ages on the temperature gradient.

**Figure supplement 2—source data 1.** Distribution of animals on thermotaxis plates at different ages.

**Figure supplement 3.** HT115-fed animals showed thermotaxis decline.

**Figure supplement 3—source data 1.** Thermotaxis decline of HT115-fed animals.

**Figure supplement 4.** AFD and AIY are alive at D5.

**Figure supplement 5.** *E. coli*-fed aged animals sense food.

**Figure supplement 5—source data 1.** Food recognition assays of young and aged animals.

**Figure supplement 6.** Aged *E. coli*-fed animals perform salt-avoidance behavior irrespective of diet.

**Figure supplement 6—source data 1.** Salt-taxis assays of young and aged animals.

conditions (*Brenner, 1974*). When the animals were placed at 17°C on a temperature gradient without food, young adults (day 1 of adulthood, D1) migrated up the temperature gradient toward 20°C (*Figure 1B*). On the other hand, aged animals (day 5 of adulthood, D5) remained around the spotted area and did not reach the area near $T_{cult}$ (*Figure 1B*), as previously reported (*Huang et al., 2020*). The ability to perform the thermotaxis behavior was quantified using the performance index, indicating the fraction of animals around $T_{cult}$ (*Figure 1A*). The performance index declined from D1 to D5 (*Figure 1C*). To further accelerate aging (*Figure 1—figure supplement 1*, *Supplementary file 1a*; *Klass, 1977*), we cultivated animals at 23°C and placed them on a temperature gradient centered at 20°C. In this condition, animals gradually lost the ability to move toward $T_{cult}$ during aging (*Figure 1—figure supplement 2*), and the performance index declined from ~0.75 at D1 to ~0.25 at D5 (*Figure 1D*). This age-dependent thermotaxis decline was not specific to OP50-fed animals because animals fed another *E. coli* strain, HT115, also showed a similar decline (*Figure 1—figure supplement 3*).

The low thermotaxis performance of *E. coli*-fed aged animals appeared to be independent of defects in motility or temperature sensation because D5 animals cultivated at 20°C could migrate down the thermal gradient relatively normally when the origin was at 23°C (*Figure 1B, C*). Consistent with this notion, we did not observe the loss of AFD or AIY neurons at D5 (*Figure 1—figure supplement 4*). Moreover, aged animals could sense food normally based on the basal slowing response in the presence of food (*Figure 1—figure supplement 5*; *Sawin et al., 2000*). The food sensation of aged animals is also reported to be normal, based on the attraction to *E. coli* (*Cornils et al., 2016*). In contrast to the basal slowing response, aged animals did not show a significantly enhanced slowing response in the starved condition (*Figure 1—figure supplement 5B*), implying that aged animals might not sense starvation normally.

To address if the defects can be observed in another associative learning behavior, we tested the salt-avoidance behavior using two assay settings (*Wicks et al., 2000*; *Saeki et al., 2001*; *Figure 1—figure supplement 6A, B*). As previously reported, naive D1 animals were attracted by NaCl, while they avoided NaCl when cultivated with NaCl in the absence of food (*Wicks et al., 2000*; *Saeki et al., 2001*; *Figure 1—figure supplement 6C, D*). In contrast to the thermotaxis behavior, D5 animals showed normal salt-avoidance behaviors (*Figure 1—figure supplement 6C, D*).

We concluded that age-dependent thermotaxis changes reflected the decline of a specific associative learning behavior and decided to use it to examine dietary effects.

## Specific LAB prevent the age-dependent thermotaxis decline

To address if bacterial diet affects the age-dependent decline in thermotaxis, we fed animals with different LAB species instead of the regular *E. coli*. We selected 35 LAB, consisting of 17 *Lactobacilli*

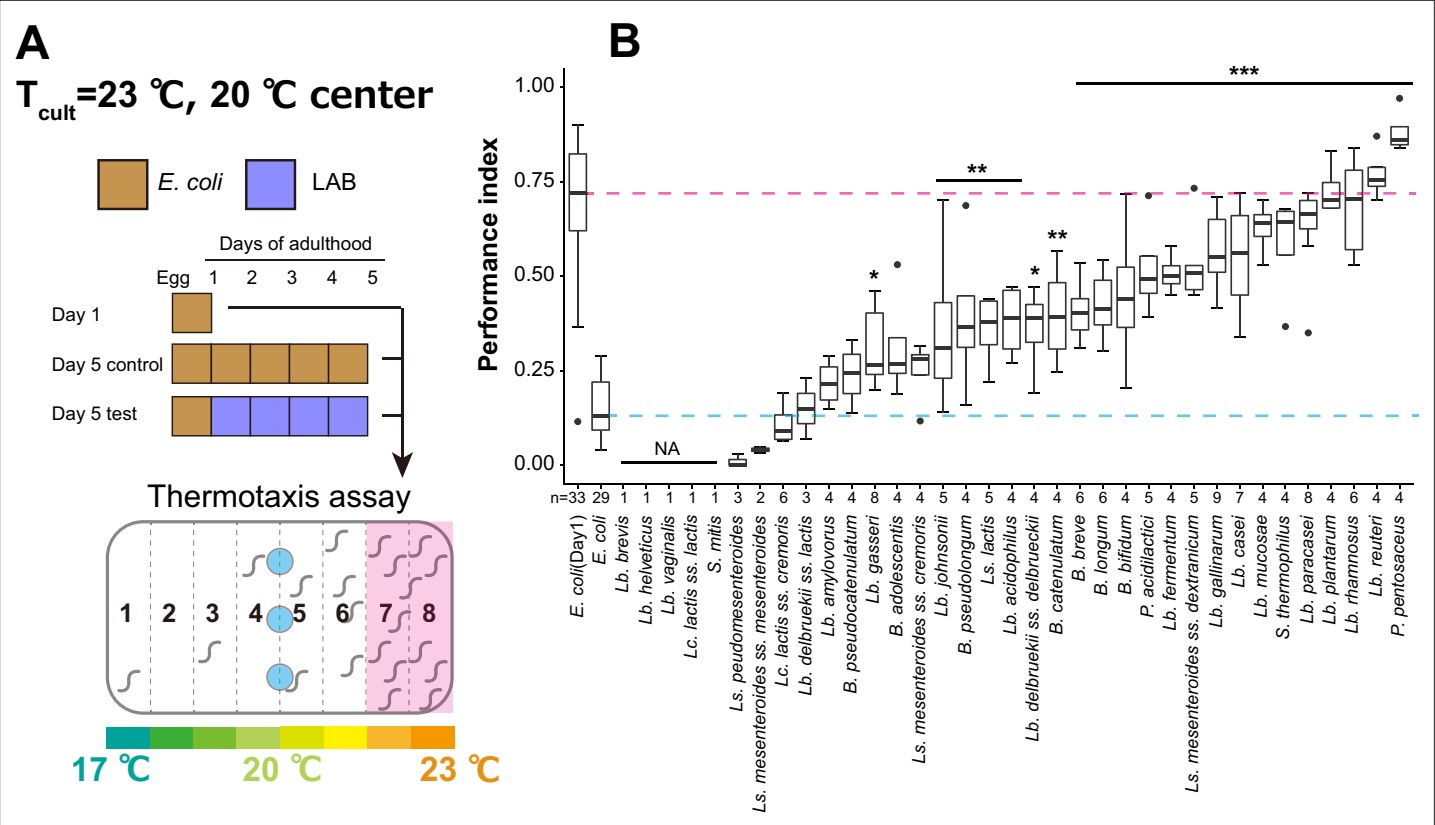

**Figure 2.** Lactic acid bacteria (LAB) screen for thermotaxis in aged animals. (**A**) Schematic of the screening procedure. Animals were cultivated at 23°C with *E. coli* until D1 and transferred to *E. coli* or LAB plates every day until D5. At D5, animals were subjected to thermotaxis assays with a thermal gradient of 17–23°C. (**B**) Box plots comparing thermotaxis performance indices of D5 animals fed LAB to those of D1 (pink dashed line) and D5 animals (light blue dashed line) fed *E. coli*. 'Not applicable' (NA) indicates that animals fed those LAB were not subjected to the assay because they were sick or dead. Abbreviations: *B*, *Bifidobacterium*; *Lb*, *Lactobacillus*; *Lc*, *Lactococcus*; *Ls*, *Leuconostoc*; *P*, *Pediococcus*; *S*, *Streptococcus*. The number of experiments is shown. Statistics: One-way analysis of variance (ANOVA) followed by Dunnett's multiple comparison test compared to D5 adults fed *E. coli*, \*\*\*p < 0.001; \*\*p < 0.01; \*p < 0.05.

The online version of this article includes the following source data and figure supplement(s) for figure 2:

**Source data 1.** Thermotaxis assays of aged animals fed with different lactic acid bacteria.

**Figure supplement 1.** Animals fed heterofermentative lactic acid bacteria (LAB) are not thermophilic.

**Figure supplement 1—source data 1.** Thermophilicity of aged animals fed with different lactic acid bacteria.

(*Lb*.), two *Pediococci* (*P*.), two *Lactococci* (*Lc*.), two *Streptococci* (*S*.), five *Leuconostoc* (*Ls*.), and seven *Bifidobacteria* (*B*.) (***Supplementary file 2***). To avoid developmental effects by feeding with LAB, we cultivated animals with *E. coli* until D1 before switching to LAB (***Figure 2A***). Animals were cultivated at 23°C and spotted at the center of the 17–23°C gradient (***Figure 2A***). Five LAB did not support the survival of animals during aging (***Figure 2B***, NA). While eight LAB did not affect the thermotaxis performance indices of the aged animals compared to *E. coli*, 22 LAB significantly increased them (***Figure 2B***). Among those, *P. pentosaceus*, *Lb. reuteri*, *Lb. rhamnosus*, and *Lb. plantarum* gave the highest performance indices (***Figure 2B***).

We first ruled out the possibility that aged animals fed the LAB were constitutively thermophilic, irrespective of the $T_{cult}$. Thermophilicity is reported for mutants of genes such as *pkc-1*/*ttx-4* encoding protein kinase C (***Okochi et al., 2005***) and *tax-6* encoding calcineurin A subunit (***Kuhara et al., 2002***). To distinguish between associative learning and thermophilicity, we shifted the thermal gradient of assay plates from 17–23 to 20–26°C for animals cultivated at 23°C (***Figure 2—figure supplement 1A***). D5 *tax-6* mutants migrated toward a higher temperature than $T_{cult}$ (***Figure 2—figure supplement 1B***). On the other hand, LAB-fed D5 animals stayed around the $T_{cult}$ (***Figure 2—figure supplement 1B***). We calculated the thermotaxis index, instead of the performance index, to quantify animals' thermal

preference (*Ito et al., 2006*; *Figure 2—figure supplement 1A*). Unlike thermophilic *tax-6* mutants, LAB-fed D5 animals showed thermotaxis indices comparable to the D1 wild type (*Figure 2—figure supplement 1C*), suggesting that LAB-fed D5 animals were not constitutively thermophilic.

We next addressed if LAB-fed D5 animals can remember a new temperature by shifting the $T_{cult}$ from 23 to 17 °C 1 day before the thermotaxis assay. D1 animals could learn the new temperature and shift their thermal preference from 23°C (*Figure 3A*) to the new $T_{cult}$, 17°C (*Figure 3B*). The thermotaxis index was also shifted accordingly (*Figure 3C*). LAB-fed D5 animals showed behavioral plasticity like D1 animals (*Figure 3B, C*). This result suggests that LAB-fed aged animals retained an ability to learn a new $T_{cult}$.

## Different LAB show various effects on lifespan and locomotion

Some LAB extend the lifespan of *C. elegans* (*Komura et al., 2013*; *Zhao et al., 2013*; *Nakagawa et al., 2016*; *Wang et al., 2020*). Therefore, better thermotaxis performance of LAB-fed aged animals might result from a systemic effect of prolonged organismal lifespan. To address this possibility, we measured the lifespan of animals fed the four LAB: *P. pentosaceus*, *Lb. reuteri*, *Lb. rhamnosus*, and *Lb. plantarum* (*Figure 4A*). To avoid the growth of *E. coli* on LAB plates after transferring animals, we used peptone-free Nematode Growth Medium (NGM) plates (*Ikeda et al., 2007*; *Lee et al., 2015*). The lack of peptone in the culture plates did not affect the dietary effects on the thermotaxis of aged animals (*Figure 4—figure supplement 1*). LAB had various effects on the lifespan of animals: *P. pentosaceus* prolonged the lifespan; *Lb. reuteri* did not affect the lifespan; *Lb. rhamnosus* and *Lb. plantarum* shortened the lifespan (*Figure 4A* and *Supplementary file 1b*).

We next examined the effect of LAB on the locomotion of aged animals using thrashing assay (*Miller et al., 1996*) and motility assay. As previously reported (*Glenn et al., 2004*; *Mulcahy et al., 2013*; *Hahm et al., 2015*), aged animals fed *E. coli* showed slight locomotion defects in both assays (*Figure 4B, C*). In the thrashing assay, *Lb. reuteri*- and *Lb. rhamnosus*-fed aged animals showed better locomotion than *E. coli*-fed aged animals, while *P. pentosaceus* and *Lb. plantarum* did not have effects (*Figure 4B*). In the motility assay, we measured the distance animals migrate on a plate. *Lb. plantarum*- and *Lb. rhamnosus*-fed aged animals showed reduced locomotion than *E. coli*-fed aged animals, while *P. pentosaceus* and *Lb. reuteri* did not have effects (*Figure 4C*). Thus, the four LAB selected based on thermotaxis had different effects on the lifespan and locomotion, implying that the dietary effect on thermotaxis is independent of lifespan and motility.

## Bacteria affect the age-dependent thermotaxis decline as nutrition

How do different bacteria affect the thermotaxis of aged animals? *C. elegans* shows different preferences in the bacterial diet (*Shtonda and Avery, 2006*). Our LAB screen used different diets during the temperature–food association before the thermotaxis assays (*Figure 2A*). It raises the possibility that the different strengths of the association during learning caused the difference in thermotaxis of aged animals. To address this issue, we switched the foods 1 day before the thermotaxis assay (*Figure 5A*). In this experiment, we used *Lb. reuteri* because it did not affect the lifespan (*Figure 4A*). Aged animals whose diet was switched from *Lb. reuteri* to *E. coli* showed the high thermotaxis performance, while aged animals with the opposite condition did not (*Figure 5A*). This result suggests that the dietary effects of thermotaxis on aged animals do not reflect the strength of association.

To examine if LAB affects animals as live bacteria or serves as nutrition, we examined the effect of bacteria heat killed at 65°C for 1 hr or at 100°C for 10 min (see Materials and methods). Like aged animals fed live bacteria, ones fed 65°C-treated *E. coli* and *Lb. reuteri* showed low- and high-performance indices in thermotaxis, respectively (*Figure 5B*). Thus, the effect of both bacteria on thermotaxis is independent of the condition of the bacteria being alive. On the other hand, animals fed 100°C-treated *E. coli* showed higher performance than those fed live *E. coli* (*Figure 5B*). This result suggests that a component in *E. coli* resistant to 65°C but sensitive to 100°C facilitates thermotaxis decline during aging. Moreover, animals fed crushed *E. coli* also showed higher thermotaxis performance, implying that the said component might be diffused after crushing (*Figure 5B*).

Next, we examined which bacteria, *E. coli* or *Lb. reuteri* has dominant effects on the thermotaxis of aged animals. We mixed *E. coli* and *Lb. reuteri* and fed animals from D1. Both *E. coli* and *Lb. reuteri* were ingested by *C. elegans* even when mixed, based on FITC labeling of bacteria (*Figure 5—figure supplement 1*). In this condition, *E. coli* had a dominant effect even when *Lb. reuteri* was mixed with

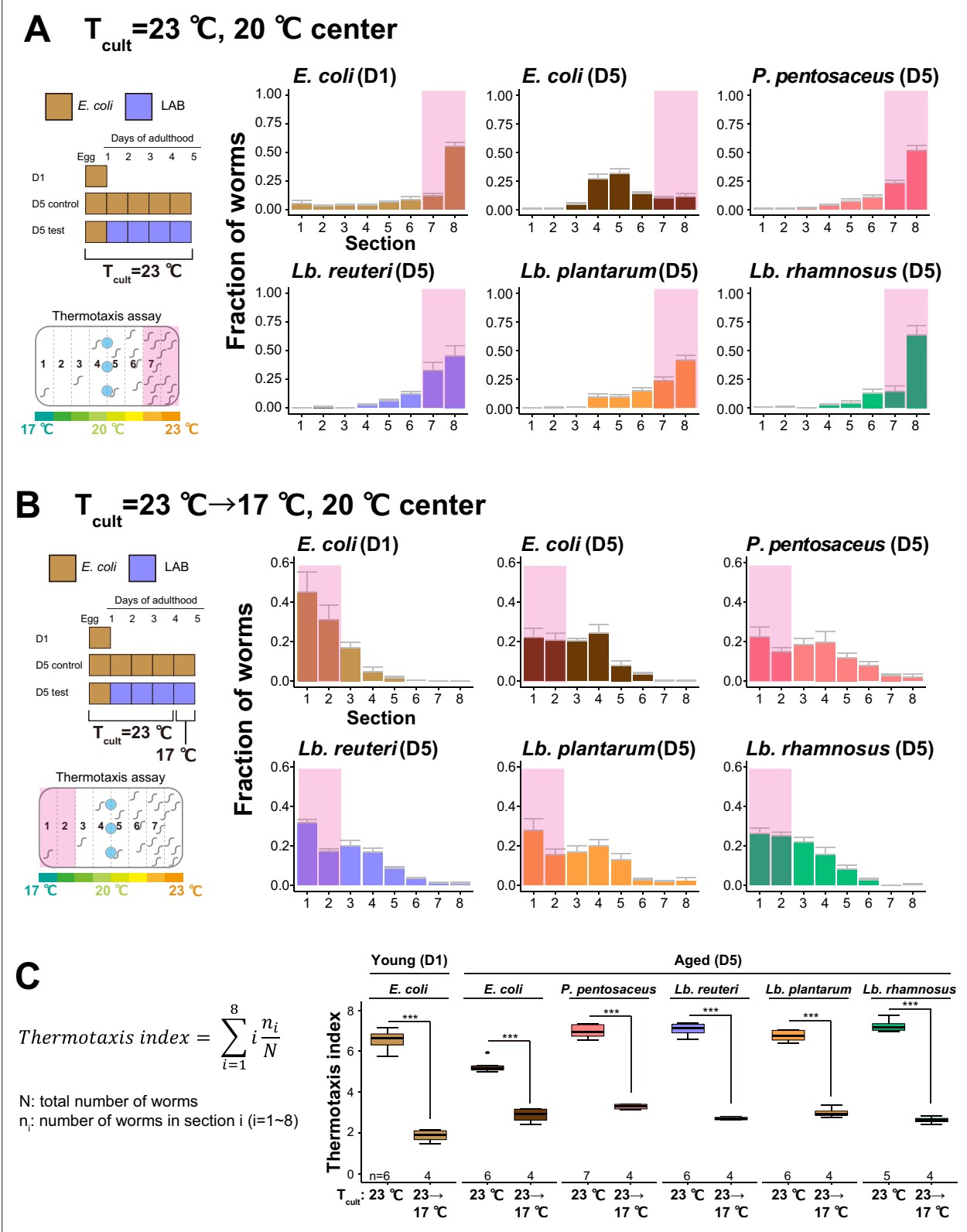

**Figure 3.** Lactic acid bacteria (LAB)-fed aged animals learn a new $T_{cult}$. (**A, B**) The distribution of animals on thermotaxis plates. Pink rectangles indicate the sections around the $T_{cult}$. (**A**) D1 or D5 animals fed indicated bacteria were cultivated at 23°C and placed at the center of a 17–23°C gradient. (**B**) Temperature shift assay. $T_{cult}$ was shifted from 23 to 17°C 1 day before the assay. Animals were placed at the center of the 17–23°C gradient. (**C**) Box plots summarizing thermotaxis indices corresponding to (**A**) and (**B**). Thermotaxis indices were calculated to examine the mean distribution of animals on

*Figure 3 continued on next page*

*Figure 3 continued*

thermotaxis plates using the indicated formula. The number of experiments is shown. Statistics: Student's *t*-test for comparison between $T_{cult}$ = 23°C and $T_{cult}$ = 23°C → 17°C, ***p < 0.001.

The online version of this article includes the following source data for figure 3:

**Source data 1.** Thermotaxis assays of aged animals fed with the select lactic acid bacteria.

**Source data 2.** Temperature shift experiment of aged animals fed with the select lactic acid bacteria.

twice as much as *E. coli* (*Figure 5C*). The dominant effect of *E. coli* appeared to require ingestion of bacteria because the exposure to the *E. coli* odor did not affect the thermotaxis ability of *Lb. reuteri*-fed aged animals (*Figure 5D*).

Collectively, we concluded that ingestion of bacterial nutrition during aging affects the thermotaxis of aged animals.

## The effects of LAB are associated with the phylogenetic tree

We explored the different features of bacteria which might affect *C. elegans* physiology. Gram-staining showed that *E. coli* was Gram-negative, while LAB were Gram-positive as expected (*Figure 6—figure supplement 1*). The morphologies and sizes of the four select LAB were different (*Figure 6—figure supplement 1*). Therefore, these physical properties of LAB may not explain the effect on the thermotaxis of aged animals.

To examine the features of the genus of LAB, we analyzed a phylogenetic tree of 35 LAB strains with the heatmap of the associated thermotaxis performance indices (*Figure 6*). This analysis revealed that LAB associated with high-performance indices were enriched in a specific clade, henceforth referred to as Clade A (*Figure 6*). Clade A, including *Lactobacilli* and *Pediococci*, was enriched in obligatory and facultatively heterofermentative species except for *P. pentosaceus*, which was obligatory homofermentative (*Figure 6*). On the other hand, the *Lactobacilli* associated with relatively low thermotaxis indices (Clade B) are all homofermentative (*Figure 6*). These results suggest that the nutritional feature of LAB shared among the clades of *Lactobacilli* might affect the thermotaxis of aged *C. elegans*.

## Neuronal *daf-16* is involved in the thermotaxis performance of *Lb. reuteri*-fed aged animals

Since the dietary effect on thermotaxis required more than 1 day to be manifested, bacterial diet seems to alter the internal state of *C. elegans*. We addressed the molecular mechanism of how different diet affects *C. elegans*. LAB can induce dietary restriction, which leads to a prolonged lifespan (*Zhao et al., 2013*). However, three out of four select LAB did not increase lifespan (*Figure 4A*). *pha-4*, an ortholog of the human FOXA2 transcription factor, is required for dietary restriction-induced longevity, and its expression is increased by dietary restriction (*Panowski et al., 2007*). In our condition, *pha-4* expression decreased in LAB-fed aged animals compared to *E. coli*-fed aged animals, suggesting that LAB-fed animals might not be under dietary restriction (*Figure 7—figure supplement 1A*). To directly test the effect of dietary restriction on the thermotaxis, we used *eat-2* mutants, which exhibit dietary restriction by defective pharyngeal pumping (*Lakowski and Hekimi, 1998*). *eat-2* mutants did not increase the performance index of *E. coli*-fed aged animals, although it may be due to the involvement of *eat-2* in thermotaxis as shown in D1 animals (*Figure 7—figure supplement 1B*). These results suggest that the high thermotaxis performance of LAB-fed aged animals is not due to dietary restriction.

To find the gene involved in the dietary effect on aged animals, we tested mutants of three genes: *nkat-1* and *kmo-1* genes that encode enzymes in the kynurenic acid-synthesizing pathway are known to be involved in butanone-associated memory in aged animals (*Vohra et al., 2018*); *daf-16* that is an ortholog of mammalian FOXO transcription factor involved in longevity (*Kenyon et al., 1993*) and LAB-dependent lifespan extension (*Grompone et al., 2012*; *Lee et al., 2015*; *Sugawara and Sakamoto, 2018*). Aged *nkat-1* and *kmo-1* mutants maintained thermotaxis ability like wild type when fed *Lb. reuteri*. On the other hand, aged *daf-16* mutants showed significantly less ability to perform thermotaxis than their D1 counterpart (*Figure 7A*); aged *daf-16* mutants fed *Lb. reuteri* distributed around a temperature slightly lower than the $T_{cult}$ (*Figure 7B*). This decreased thermotaxis ability in

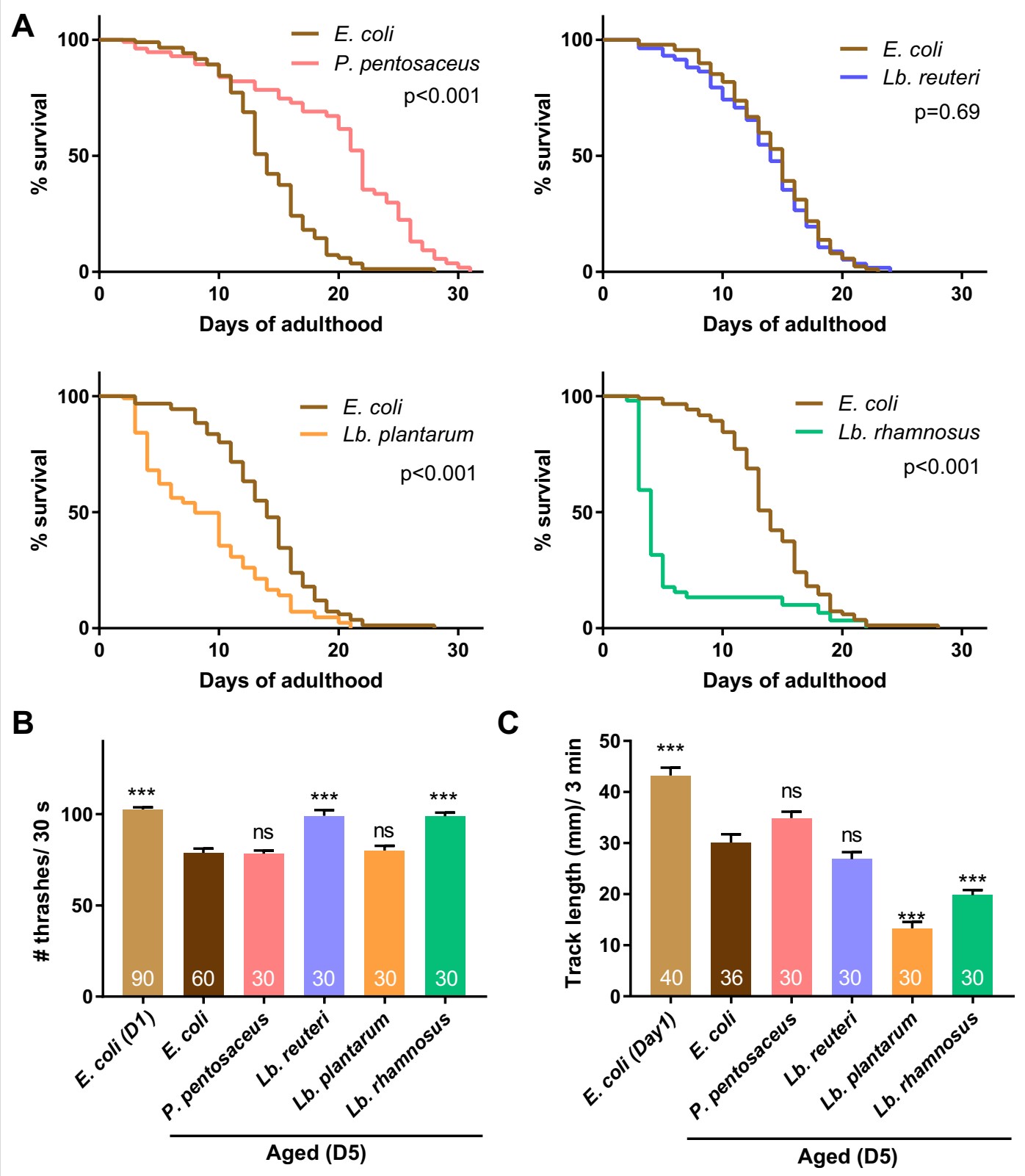

**Figure 4.** Lactic acid bacteria (LAB) show various effects on lifespan and locomotion. Animals were fed *E. coli* until D1 and indicated bacteria after D1. (**A**) Survival curves of animals fed indicated LAB are shown with control animals fed *E. coli*. Nematode Growth Medium (NGM) plates without peptone were used to avoid the undesired growth of *E. coli* on LAB plates. *N* = 4 experiments with 25 animals/experiment (100 animals in total). Statistics: Log-rank test. p values are shown. The number of thrashes in liquid (**B**) and distance of migration in 3 min on plates with *E. coli* (**C**) were measured to

*Figure 4 continued on next page*

*Figure 4 continued*

examine the locomotion of aged animals. The number of animals is shown in bars. Error bars: Standard error of the mean (SEM). Statistics: One-way analysis of variance (ANOVA) followed by Dunnett's multiple comparison test compared to D5 fed *E. coli*, ***p < 0.001; ns, p > 0.05.

The online version of this article includes the following source data and figure supplement(s) for figure 4:

**Source data 1.** Lifespan of animals fed with the select lactic acid bacteria.

**Source data 2.** Thrashing assays of aged animals fed with the select lactic acid bacteria.

**Source data 3.** Locomotion of aged animals fed with different lactic acid bacteria.

**Figure supplement 1.** Diet affects thermotaxis of aged animals on peptone-free plates.

**Figure supplement 1—source data 1.** Thermotaxis assays using peptone-free plates.

aged *daf-16* mutants fed *Lb. reuteri* was not due to shortened lifespan because *daf-16* mutants had comparable lifespan to wild-type animals when fed *Lb. reuteri* (*Figure 7C*, *Supplementary file 1c*). DAF-16 is known to be activated in *daf-2* mutants. Therefore, we speculated that *daf-2* mutants might show high thermotaxis performance even in *E. coli*-fed aged animals. However, *daf-2* was required for normal thermotaxis in both young and aged animals (*Figure 7D*). *daf-16* possesses several isoforms with different expression patterns and functions (*Figure 8A*; *Kwon et al., 2010*). The b isoform is a neuronal isoform involved in AIY development (*Christensen et al., 2011*). The other isoforms are more broadly expressed in most tissues, including neurons. We addressed which isoform is necessary for the effect on thermotaxis of aged animals. *daf-16(mg54)* has a single-nucleotide polymorphism which introduces an amber stop codon mutation and affects the exons of all *daf-16* isoforms except the b isoform (*Figure 8A*; *Ogg et al., 1997*). To knock out only the b isoform, we generated *daf-16(knj36)* which introduced 7 bp deletion in the first exon of the b isoform, located in the intron of the other isoforms (*Figure 8A*). Both *daf-16(mg54)* and *daf-16(knj36)* did not affect the dietary effects on the thermotaxis of aged animals (*Figure 8B*). This result implies that the b isoform and other isoforms complement each other and that having either one is sufficient to give the dietary effect. Consistently, the expression of *daf-16b* under the control of its own promoter rescued the deletion mutants, *daf-16(mu86)*, confirming the sufficiency of the *daf-16* b isoform.

We addressed in which tissue *daf-16* functions. Given that the expression of either b isoform or non-b isoforms was sufficient to provide the dietary effects (*Figure 8B*), *daf-16* functions in a tissue where both b and non-b isoforms are expressed. We thus focused on pharyngeal muscles, body wall muscles, and neurons (*Nagashima et al., 2019*). Expression of *daf-16b* in pharyngeal muscles or body wall muscles did not rescue *daf-16(mu86)* (*Figure 8C*). On the other hand, the *daf-16* b isoform expressed in all neurons rescued the low performance of *Lb. reuteri*-fed aged *daf-16(mu86)* mutants (*Figure 8C*). This result suggests that *daf-16* functioning in neurons is involved in the high thermotaxis performance of *Lb. reuteri*-fed aged animals.

Since *daf-16* is involved in the development of neurons including AIY, which is important for thermotaxis (*Christensen et al., 2011*), it is possible that *daf-16* plays a role during development instead of during aging. To address this issue we knocked down *daf-16* in a time-specific manner using the Auxin-Inducible Degron (AID) system (*Nishimura et al., 2009*; *Zhang et al., 2022*). In the AID system, a target protein with a degron tag is degraded in an auxin-dependent manner in tissues expressing TIR1. We first confirmed that the AID system with degron-tagged *daf-16* phenocopied *daf-16(mu86)* phenotype (*Figure 8—figure supplement 1*). Then, we asked the critical period of *daf-16* requirement and found that *daf-16* is required during aging instead of development to maintain the thermotaxis performance of *Lb. reuteri*-fed aged animals (*Figure 8D*).

## Diet and age affect DAF-16 target genes

To comprehensively understand the dietary effect on aging, we carried out RNA sequencing of eight samples: D1, D5 fed *E. coli*, D5 fed homo-fermentative LAB, which gave low thermotaxis index (*Lb. gasseri* and *Lb. delbrueckii*), and D5 fed heterofermentative LAB, which gave high thermotaxis index (*P. pentosaceus*, *Lb. reuteri*, *Lb. rhamnosus*, and *Lb. plantarum*) (*Figure 9A*, *Supplementary file 3a*). The heatmap for the differentially expressed genes is shown in *Figure 9—figure supplement 1*. Principal component analysis of transcriptome data revealed that the D5 fed LAB clustered together separately from D1 and D5 fed *E. coli*, irrespective of the fermentation mode of LAB (*Figure 9B*). The first principal component (PC1) explained 76% of the entire variance and appeared

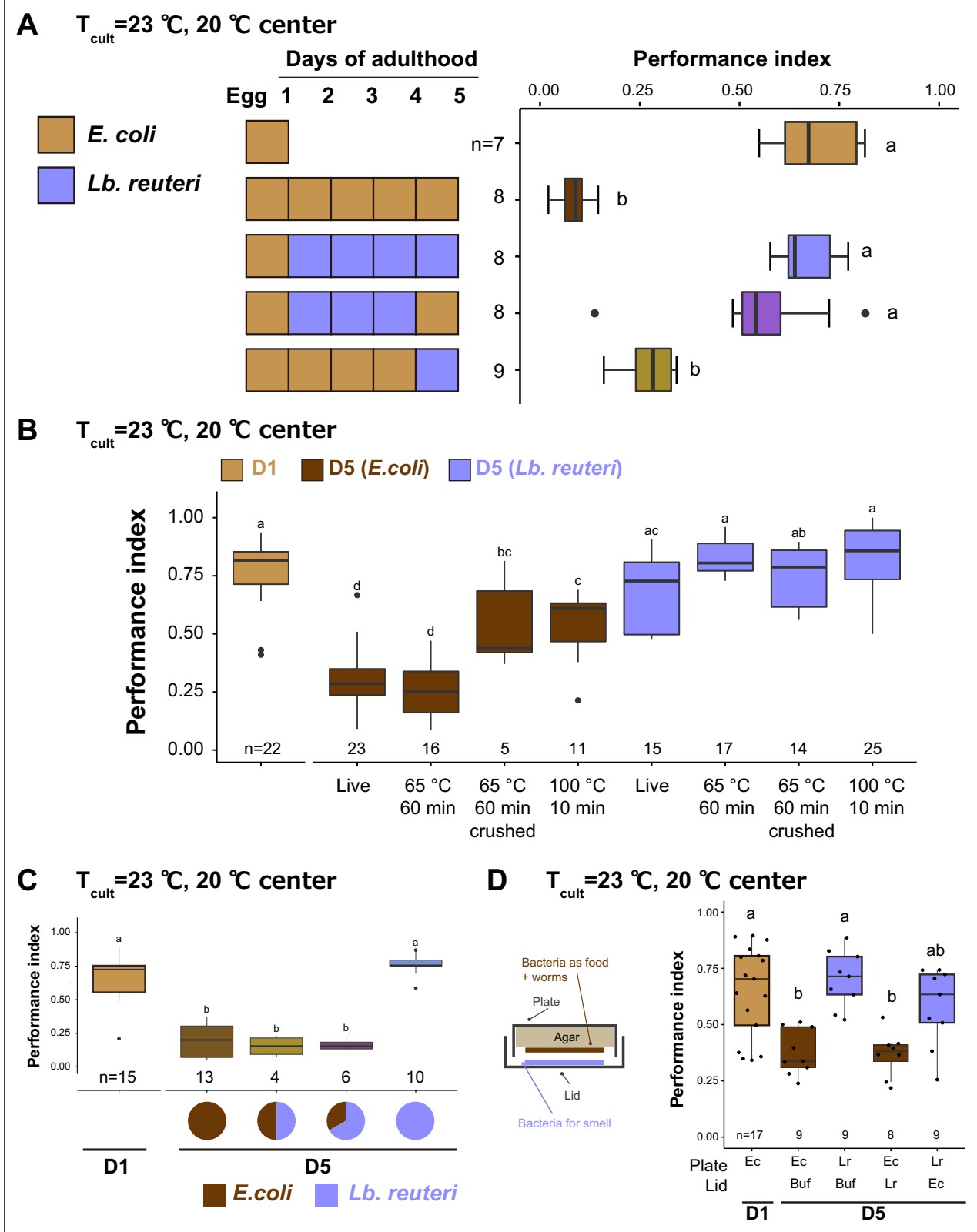

**Figure 5.** Bacteria affect thermotaxis of aged animals as nutrition. Box plots show thermotaxis performance indices of animals fed indicated bacteria and cultivated at 23°C. Aged animals were transferred every day to new plates from D1. The number of experiments is shown. Statistics: The mean indices marked with distinct alphabets are significantly different (p < 0.05) according to one-way analysis of variance (ANOVA) followed by Tukey–Kramer test. (**A**) The short-term effects of diet. The diet was switched 1 day before the thermotaxis assay, as indicated in the schematic. (**B**) The effect of heat

*Figure 5 continued on next page*

*Figure 5 continued*

treatment and crushing of bacteria. The bacteria were killed by incubating at 65°C for 1 hr or 100°C for 10 min. After heat treatment, bacteria were crushed using a bead-based homogenizer for the crushed condition. (**C**) The mixture of bacteria. Live *E. coli* and *Lb. reuteri* were mixed at a 1:1 or 1:2 ratio with the final concentration of 0.1 g/ml and used as a diet. (**D**) The effect of bacterial odor. Animals were exposed to the bacterial odor by putting the bacterial solution on the lid and cultivated, as shown in the schematic. Ec: *E. coli*, Lr: *Lb. reuteri*.

The online version of this article includes the following source data and figure supplement(s) for figure 5:

**Source data 1.** The effect of switching bacteria on the thermotaxis behavior of aged animals.

**Source data 2.** The effect of heat-killed bacteria on the thermotaxis behavior of aged animals.

**Source data 3.** The effect of mixed bacteria on the thermotaxis behavior of aged animals.

**Source data 4.** The effect of bacterial smell on the thermotaxis behavior of aged animals.

**Figure supplement 1.** Animals ingest *E.coli* and *Lb. reuteri*.

to represent the difference in age irrespective of diet (*Figure 9B*, PC1). To characterize what genes contributed to PC1, we selected the top 5% genes positively or negatively correlating to PC1 (666 genes, each, among 13331 genes in total, *Figure 9—figure supplement 2A*, *Supplementary file 3b, c*) and performed Gene Ontology (GO) analysis using Metascape (*Zhou et al., 2019b*). The genes correlating to PC1 were enriched with the categories such as oogenesis (GO:0048477; p = 1.0 × 10$^{-10}$; Enrichment, 5.7) and structural constituent of cuticle (GO:0042302; p = 1.0 × 10$^{-34}$; Enrichment, 8.3) (*Figure 9—figure supplement 2C*, *Supplementary file 3d, e*). The second principal

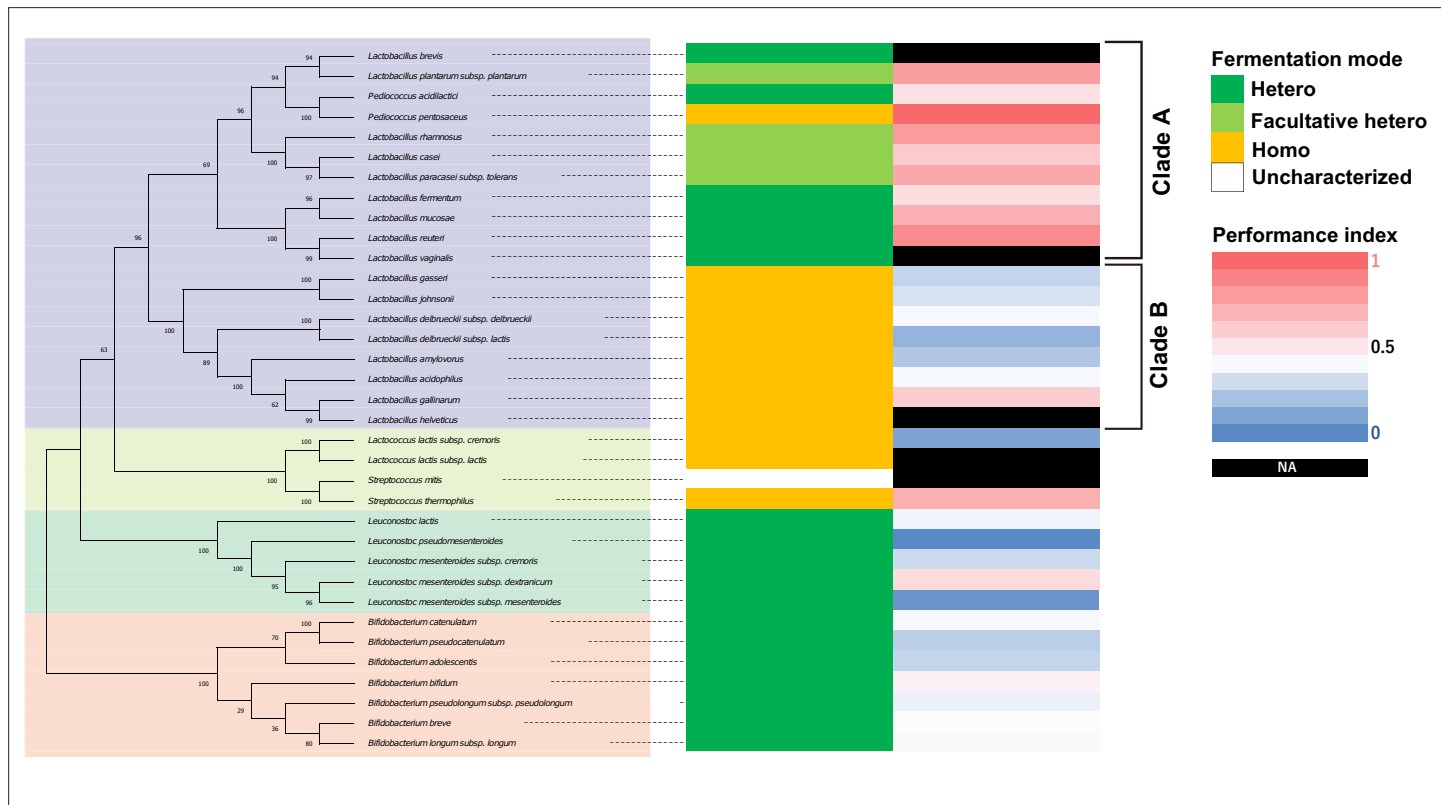

**Figure 6.** *Lactobacilli* in a clade are associated with high thermotaxis performance of aged animals. Phylogenetic tree of lactic acid bacteria (LAB) based on 16S rRNA is shown with fermentation mode, and heatmap of performance indices of aged animals fed indicated LAB from D1. Bootstrap values are indicated at each node on the phylogenetic tree. Fermentation modes were categorized based on previous studies (see Table S1*Supplementary file 2*). The same data as *Figure 2B* were used for thermotaxis performance indices. NA in the performance indices heatmap indicates that animals fed those LAB were not subjected to the thermotaxis assay because they were sick or dead. Fermentation mode indicates obligatory hetero- (green), facultatively hetero- (light green), and obligatory homofermentative (orange) LAB.

The online version of this article includes the following figure supplement(s) for figure 6:

**Figure supplement 1.** Images of Gram-stained bacteria.

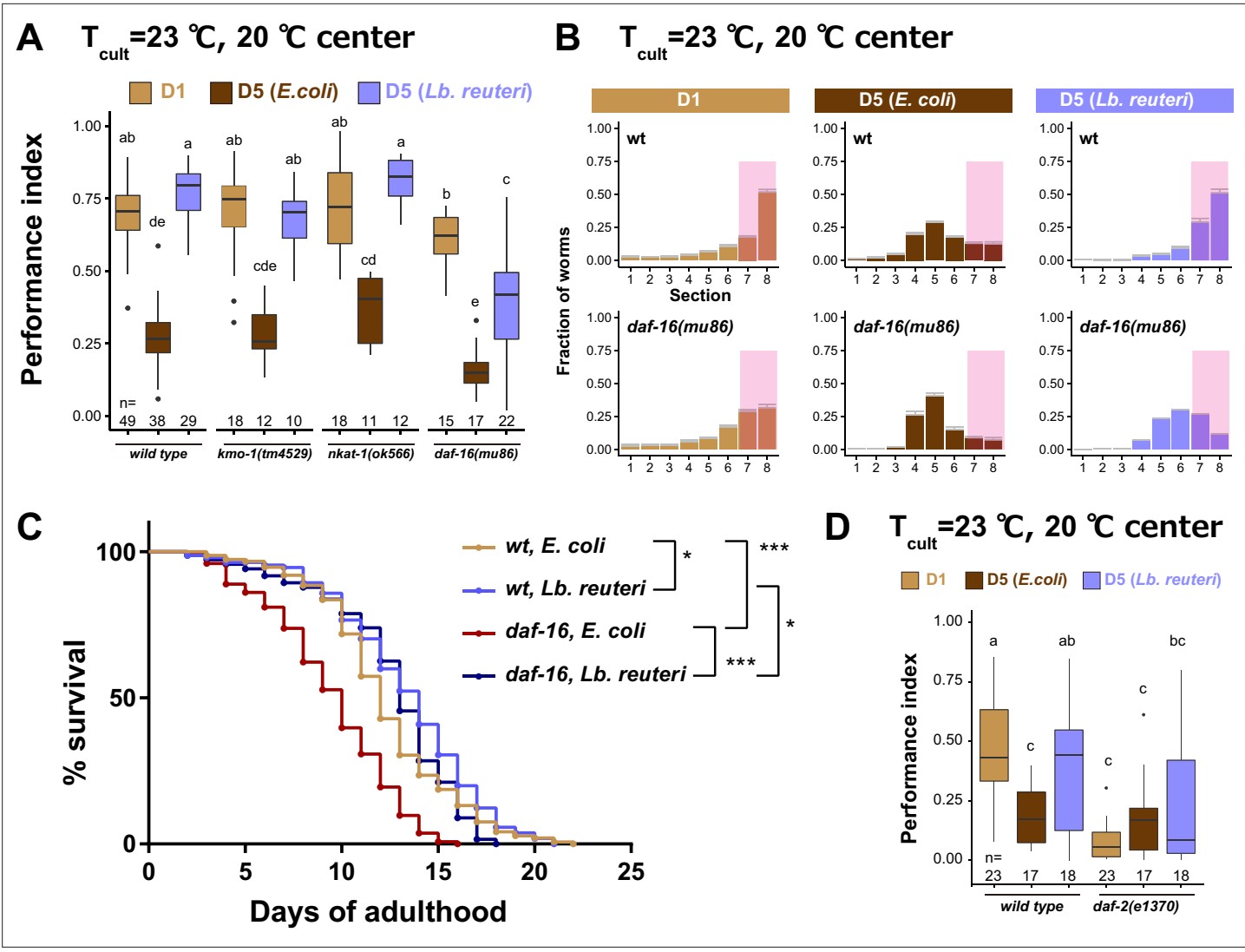

**Figure 7.** *daf-16* is involved in the effect of *Lb. reuteri* on thermotaxis in aged animals. (**A**) Box plots summarizing thermotaxis performance indices of animals with indicated genotypes in the different diet and age conditions. Animals were cultivated at 23°C with *E. coli* or *Lb. reuteri* from D1. The number of experiments is shown. Statistics: The mean indices marked with distinct alphabets are significantly different (p < 0.05) according to two-way analysis of variance (ANOVA) followed by Tukey–Kramer test. (**B**) Distribution of animals of indicated conditions on thermotaxis plates. Pink rectangles indicate the sections around the $T_{cult}$. (**C**) Survival curves of animals with indicated genotypes fed *E. coli* or *Lb. reuteri* from D1 and cultivated at 23°C. Nematode Growth Medium (NGM) plates without peptone were used. N = 6 experiments with 25 animals/experiment (150 animals in total). Statistics: Log-rank test, ***p < 0.001; *p < 0.05. (**D**) Box plots summarizing thermotaxis performance indices of animals with indicated genotypes. Animals were cultivated at 15°C for 96 hr to avoid dauer formation of *daf-2(e1370)* and then incubated at 23°C until D1 or D5 with *E. coli* or *Lb. reuteri*. The number of experiments is shown. Statistics: The mean indices marked with distinct alphabets are significantly different (p < 0.05) according to two-way ANOVA followed by Tukey–Kramer test.

The online version of this article includes the following source data and figure supplement(s) for figure 7:

**Source data 1.** Thermotaxis of kmo-1, nkat-1, and daf-16 mutants.

**Source data 2.** Distributions of daf-16 mutants on thermotaxis plates.

**Source data 3.** Lifespan of daf-16 mutants fed with *E. coli* or *Lb. reuteri*.

**Source data 4.** Thermotaix of daf-2 mutants.

**Figure supplement 1.** Lactic acid bacteria (LAB)-fed animals are not dietary restricted.

**Figure supplement 1—source data 1.** pha-4 expression of aged animals fed with the select lactic acid bacteria.

**Figure supplement 1—source data 2.** Thermotaxis assays of eat-2 mutants.

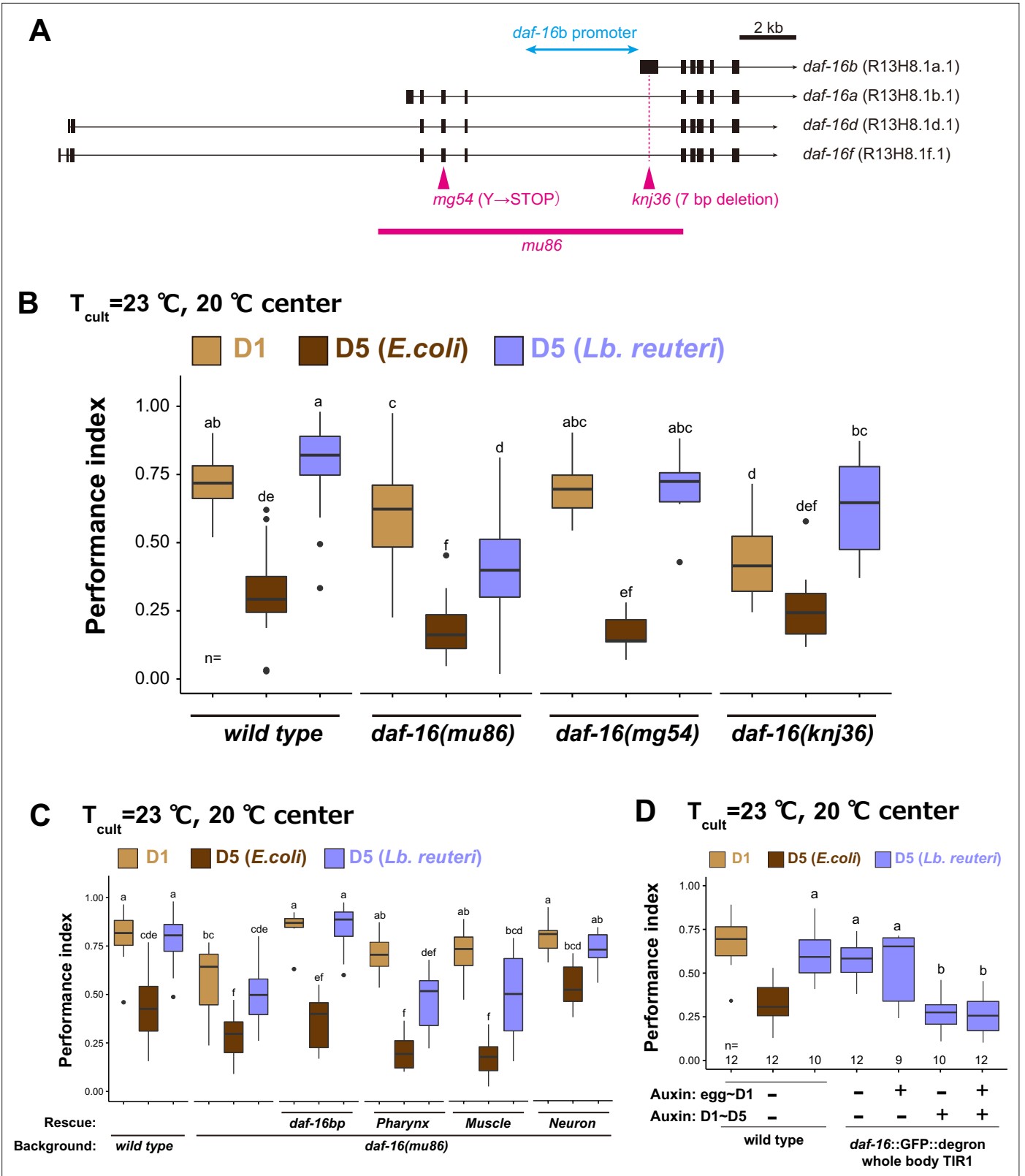

**Figure 8.** *daf-16* b isoform functions in neurons to maintain high thermotaxis ability in aged animals fed *Lb. reuteri*. (**A**) Schematic of *daf-16* locus with representative isoforms based on WormBase (https://wormbase.org). Black boxes and black lines indicate exons and introns, respectively. Arrows indicate 3'UTR. Alleles used in this study are shown in magenta. The promoter of the *daf-16* b isoform (4.9 kbp) is indicated in light blue. (**B, C**) Box plots summarizing thermotaxis performance indices of animals with indicated genotypes in the different age and diet conditions. Animals were cultivated

*Figure 8 continued on next page*

*Figure 8 continued*

at 23°C with *E. coli* or *Lb. reuteri* from D1. The number of experiments is shown. Statistics: The mean indices marked with distinct alphabets are significantly different (p < 0.05) according to two-way analysis of variance (ANOVA) followed by Tukey–Kramer test. (**B**) Analysis of different alleles of *daf-16* indicated in (**A**). (**C**) Tissue-specific rescue of *daf-16*. The single-copy insertions of *daf-16b* fragment with introns under tissue-specific promoters were used to examine if it rescues *daf-16(mu86)*: *myo-2*p, pharynx; *myo-3*p, body-wall muscle; *rgef-1*p, pan-neuron. (**D**) Time-specific knock down of *daf-16*. *daf-16*::degron animals carrying TIR1 expressed in the whole body were treated with auxin during development and/or aging.

The online version of this article includes the following source data and figure supplement(s) for figure 8:

**Source data 1.** The effect of different daf-16 alleles on the thermotaxis behavior.

**Source data 2.** Tissue-specific daf-16 rescue of the thermotaxis behavior.

**Source data 3.** The effect of timing-specific knockdown of daf-16 on the thermotaxis behavior.

**Figure supplement 1.** Auxin and Auxin-Inducible Degron (AID) tag do not affect the thermotaxis.

**Figure supplement 1—source data 1.** The effect of daf-16 knockdown using AID on the thermotaxis behavior.

component (PC2) explained 9% of the entire variance and appeared to represent the difference between *E. coli* and LAB (*Figure 9B*, *Supplementary file 3f, g*). GO analysis showed that the genes correlating to PC2 were enriched with the categories such as glucuronosyltransferase activity (GO:0015020; p = $1.0 \times 10^{-10}$; Enrichment, 6.7) and biological processes involved in interspecies interaction between organisms (GO:0044419; p = $1.0 \times 10^{-34}$; Enrichment, 5.3) (*Figure 9—figure supplement 2B, D*, *Supplementary file 3h, i*).

Among these categories, we found the neuropeptide signaling pathway (GO:0007218; p = 5.1 × $10^{-4}$; Enrichment, 2.9) particularly intriguing because it can work as signals from the intestine to neurons. To investigate this possibility, we analyzed the mutants of four proprotein convertases which are crucial for neuropeptide synthesis (*Husson et al., 2006*; *Li and Kim, 2008*). Among those, *kpc-1* and *egl-3* were required for thermotaxis of D1 animals; *bli-4* showed similar phenotypes to the wild type irrespective of age or diet (*Figure 9C*). On the other hand, *aex-5* mutants showed higher thermotaxis ability in the *E. coli*-fed D5 condition, suggesting that *aex-5* was required to decrease the thermotaxis ability during aging (*Figure 9C*).

The principal component analysis revealed the feature of gene expression of animals of different ages and diet. However, it did not explain the difference in thermotaxis ability in (1) D5 animals fed *E. coli* or homofermentative LAB and (2) D5 animals fed hetero fermentative LAB. Therefore, we investigated differentially expressed genes between these two groups (*Figure 9A*). We found 65 genes whose expression is >2 times higher in Group 1 (*E. coli* and homofermentative LAB) than in Group 2 (heterofermentative LAB) (*Supplementary file 3j*). These genes included fungus-induced protein-related genes (*fipr-22*, *fipr-23*, and *fipr-24*) (*Figure 9D*) and were enriched with the category such as regulation of lipid localization (GO:1905952; p = $1.3 \times 10^{-4}$; Enrichment, 29.0) (*Figure 9E*, *Supplementary file 3k*). On the other hand, we found 71 genes whose expression is >2 times higher in Group 2 than in Group 1 (*Supplementary file 3l*). These genes included lysozyme genes (*lys-5* and *lys-6*) (*Figure 9D*) and were enriched with the category such as regulation of defense response to other organisms (GO:0098542; p = $2.0 \times 10^{-5}$; Enrichment, 6.6) (*Figure 9E*, *Supplementary file 3m*).

Since the high thermotaxis ability of *Lb. reuteri*-fed aged animals were *daf-16* dependent, we investigated the relevance of *daf-16* and RNAseq data. The expression of *daf-16* itself was not changed in the RNAseq (*Supplementary file 3a*). This result implied that DAF-16 protein might be activated in aged animals fed *E. coli* or homofermentative LAB. We then investigated the enrichment of *daf-16* targets in the whole body (1663 upregulated genes and 1733 downregulated genes; *Tepper et al., 2013*). The genes highly expressed in aged animals fed *E. coli* or homofermentative LAB were significantly enriched with upregulated genes by DAF-16 (Normalized Enrichment Score [NES], −1.34; False Discovery Rate [FDR], 0.11; p < 0.001) (*Figure 9F*). On the other hand, the genes highly expressed in heterofermentative LAB were significantly enriched with downregulated genes by DAF-16 (NES, 1.53; FDR, 0.046; p < 0.001) (*Figure 9F*). Interestingly, this is the opposite of the prediction based on the *daf-16* dependency of the thermotaxis of *Lb. reuteri*-fed animals. Namely, *daf-16* is predicted to be activated in heterofermentative *Lb. reuteri*-fed aged animals. Therefore, *Lb. reuteri* might activate DAF-16 in a tissue and/or target-specific manner. Consistent with this notion, the general activation of DAF-16 using *daf-2* mutants did not improve the thermotaxis ability of aged animals (*Figure 7D*).

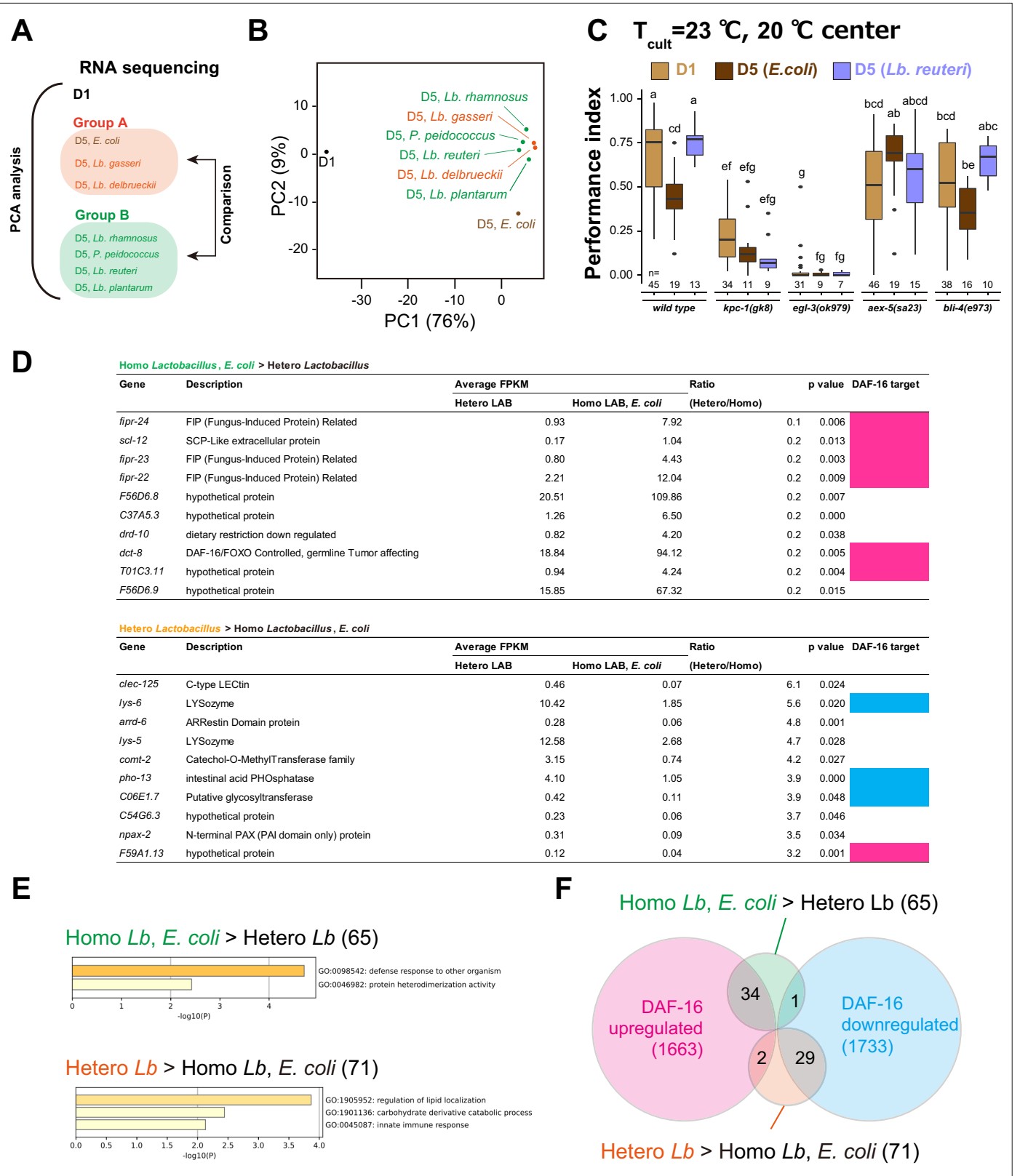

**Figure 9.** Transcriptome analysis on the effect of aging and diet. (**A**) Schematic of transcriptome analysis. We carried out RNA sequencing of indicated samples (eight in total). Principal component analysis was carried out using all samples. Differentially expressed genes were analyzed using D5 samples to examine the difference between Groups A and B. (**B**) A scatter plots of the two variables projected on the first and second principal components. The percentage of variance explained by each principal component is indicated in brackets. (**C**) Mutants of the genes encoding proprotein convertases.

*Figure 9 continued on next page*

*Figure 9 continued*

Box plots summarizing thermotaxis performance indices of animals with indicated genotypes in the different age and diet conditions. Animals were cultivated at 23°C with *E. coli* or *Lb. reuteri* from D1. The number of experiments is shown. Statistics: The mean indices marked with distinct alphabets are significantly different (p < 0.05) according to two-way analysis of variance (ANOVA) followed by Tukey–Kramer test. (**D**) Differentially expressed genes (DEG) between Groups A and B in (**A**). DEG were ranked by the ratio between Group A's and B's average expression, and the top 10 genes are shown. Magenta and light blue indicate genes that are up- and downregulated by DAF-16 (**Tepper et al., 2013**), respectively. (**E**) Gene ontology analysis of differentially expressed genes between Groups A and B. *x*-Axis indicates log₁₀[p-value of enrichment]. The number of genes is indicated in brackets. (**F**) Venn diagram showing the overlap between differentially expressed genes of our samples and DAF-16 targets (**Tepper et al., 2013**). The number of genes is indicated in brackets.

The online version of this article includes the following source data and figure supplement(s) for figure 9:

**Source data 1.** Thermotaxis assays of proprotein convertase mutants.

**Source data 2.** List of genes enriched in aged animals fed with different bacteria.

**Figure supplement 1.** Heatmap for differentially expressed genes.

**Figure supplement 2.** Principal component analysis of the transcriptome data of the animals of different ages and diet.

**Figure supplement 2—source data 1.** Contribution of each gene on PC1.

**Figure supplement 2—source data 2.** Contribution of each gene on PC2.

## Discussion

Addressing the causal relationship between diet and their effects on animals' physiology is challenging in human or mammalian models because microbiota in the gut and diet are complex. It is especially true in the context of aging because of their long lifespan. Using *C. elegans* as a model, we provide evidence of the dietary effect on the age-dependent behavioral decline discernible from the lifespan.

### Aging and diet show various effects on different behaviors and organismal lifespan

This study demonstrated that diet affects the age-dependent decline of thermotaxis behavior. We ruled out the possibility that the high performance of LAB-fed aged animals was due to thermophilicity, stronger association to LAB, better motility, dietary restriction, or longer lifespan. Thus, aging and diet likely affect the thermosensory circuit. The primary thermosensory neuron AFD (**Mori and Ohshima, 1995**) cell-autonomously stores temperature memory (**Kobayashi et al., 2016**). Although the Ca²⁺ response in AFD is reported to be defective in aged animals (**Huang et al., 2020**), the temperature sensation itself does not seem to be abolished in aged animals because they could migrate down the gradient (**Figure 1B, C**). AFD thermosensory neurons synapse onto and regulate AIY interneurons by switching excitatory and inhibitory signals in a context-dependent manner (**White et al., 1986**; **Mori and Ohshima, 1995**; **Nakano et al., 2020**). AIY neurons are reported to be a site of action of *age-1* PI3 kinase, which is upstream of *daf-16* in isothermal tracking behavior (**Murakami et al., 2005**). Therefore, aging and diet might affect AIY interneurons.

The degree of age-dependent decline seems to depend on behaviors. *E. coli*-fed animals experienced an age-dependent decline in thermotaxis but not in salt-avoidance behavior. Despite the similar effects of *P. pentosaceus*, *Lb. reuteri*, *Lb plantarum*, and *Lb. rhamnosus* on thermotaxis in aged animals, these LAB showed various effects on locomotion. In thermotaxis, aged animals showed more severe defects in migrating up the thermal gradient than migrating down (**Figure 1B, C**). Thermotaxis is achieved by multiple steps: sensing temperature, recognizing food, associating food and temperature, memorizing $T_{cult}$, and migrating toward $T_{cult}$ (**Kimata et al., 2012**; **Aoki and Mori, 2015**; **Goodman and Sengupta, 2018**). Thus, the different severities of thermotaxis decline between migration up and down the gradient in aged animals might be attributed to the different neural circuits responsible for those conditions, as previously reported (**Ikeda et al., 2020**).

Neuronal aging is discernible from organismal lifespan. *nkat-1* mutants prevent age-dependent memory decline in associative learning between food and butanone without changing lifespan (**Vohra et al., 2018**). Similarly, we found that *Lb. reuteri* improved thermotaxis in aged animals without changing their lifespan. More strikingly, *Lb plantarum* and *Lb rhamnosus* shortened the lifespan while they had beneficial effects on the thermotaxis of D5 adults. This different dietary condition will allow us to address the mechanism underlying phenotypic variation in aged animals independent from organismal lifespan and genetic perturbation.

## Diet affects age-dependent thermotaxis decline as nutrients

Previous reports elucidated how bacterial diet affects *C. elegans* as nutritional components, gut microbiota, and/or pathogen (*Kumar et al., 2020*; *Zhou et al., 2019a*) probably by changing *C. elegans* metabolites (*Reinke et al., 2010*; *Gao et al., 2017*) and gene expression (*MacNeil et al., 2013*).

Both live *E. coli* and LAB can colonize animals (*Portal-Celhay et al., 2012*; *Berg et al., 2016*; *Chelliah et al., 2018*; *Park et al., 2018*). Live bacteria are necessary for some physiological roles; secreted enterobactin from live *E. coli* in the gut promotes *C. elegans* growth (*Qi and Han, 2018*); live, but not dead, LAB reduces the susceptibility to pathogenic bacteria *Pseudomonas aeruginosa*. On the other hand, live bacteria are unnecessary in different contexts; heat-killed *Lb. paracasei* and *Bifidobacterium longum* extend *C. elegans* lifespan (*Sugawara and Sakamoto, 2018*; *Wang et al., 2020*). In our thermotaxis assay on aged animals, *E. coli* and LAB killed by 65°C treatment had similar effects to live bacteria. This result implies that, instead of the action of live bacteria, such as pathogenic effects of *E. coli* (*Cabreiro and Gems, 2013*), bacterial nutrition might be responsible for the effect on the thermotaxis of aged *C. elegans*. Feeding mixed bacteria suggests that *E. coli* has a dominant effect over *Lb. reuteri* to reduce the thermotaxis ability of aged animals. *E. coli* appeared to have components with a negative effect on thermotaxis of aged animals, and these components are vulnerable at 100°C and diffusible after crushing. This notion is further supported by the fact that *E. coli*'s negative effect was ameliorated by the genetic mutant of *aex-5*. Since the smell of bacteria did not reverse the effect of diet, the ingestion of bacterial nutrition causes the dietary effects. Metabolites in bacterial diet affect *C. elegans* physiology; some metabolites are beneficial, while others are toxic (*Zhou et al., 2019a*). Coenzyme Q in *E. coli* shortens the lifespan of *C. elegans* (*Larsen and Clarke, 2002*). Bacterial nitric oxide and folate are also positive and negative regulators of *C. elegans* lifespan, respectively (*Virk et al., 2012*; *Gusarov et al., 2013*). Vitamin 12 in *Comamonas aquatica* accelerates development and reduces fertility without changing lifespan (*Watson et al., 2014*). Given that different metabolites are produced by different LAB (*Tomita et al., 2017*), these metabolites might be responsible for the different effects on the thermotaxis of aged *C. elegans*.

Our results indicated that LAB that gave high-performance indices of thermotaxis are associated with a clade enriched in heterofermentative *Lactobacilli* and *Pediococci* (Clade A in *Figure 6*). Heterofermentative LAB produce not only lactic acid and ATP but also several other end products such as ethanol and $CO_2$ from glucose. On the other hand, homofermentative LAB converts glucose into two molecules of lactic acid and ATP. Heterolactic fermentation itself does not explain the high-performance index in thermotaxis of aged animals because heterofermentative *Leuconostoc* and *Bifidobacteria* species did not give the high-performance indices. Metabolites other than lactic acid, ethanol, and $CO_2$ also differ between hetero- and homofermentative *Lactobacilli* (*Tomita et al., 2017*). Metabolites enriched in heterofermentative *Lactobacilli* include a neurotransmitter GABA and tyramine, a substrate to synthesize neurotransmitter octopamine; metabolites enriched in homofermentative *Lactobacilli* include 4-hydroxyphenyllactic acid and acetoin. We note that Tomita et al. reported the metabolites in the media (*Tomita et al., 2017*) while we supply bacteria to animals after washing off the bacterial media. Nonetheless, metabolites enriched in different *Lactobacilli* might affect the age-dependent thermotaxis decline.

## Bacterial diet modulates genetic pathways in *C. elegans* neurons

LAB can extend the lifespan of *C. elegans* either by dietary restriction-dependent (*Zhao et al., 2013*) or -independent mechanisms (*Komura et al., 2013*; *Nakagawa et al., 2016*). The mechanism underlying the dietary effect on the thermotaxis decline does not seem to depend on the activation of the dietary restriction pathway. First, the expression of *pha-4* was low. Second, the lifespan of LAB-fed animals was not necessarily prolonged. Third, *eat-2* mutants, which mimic dietary restriction, did not improve thermotaxis in aged animals fed *E. coli*. Fourth, *kmo-1* and *nkat-1* genes involved in dietary restriction-dependent beneficial effects on associative learning (*Vohra et al., 2017*) did not affect the dietary effects on thermotaxis of aged animals. Fifth, no correlation was observed between the thermotaxis ability of aged animals and body size or fat accumulation.

Different LAB activate distinct genetic pathways such as IIS pathway important for lifespan regulation and p38 mitogen-activated protein kinase (MAPK) pathway important for innate immunity. *Lb. rhamnosus* and *B. longum* extend the lifespan of *C. elegans* by modulating the IIS pathway consisting of DAF-2 and DAF-16 (*Grompone et al., 2012*; *Sugawara and Sakamoto, 2018*). *B. infantis* extends

the lifespan of *C. elegans* via the PMK-1 p38 MAPK pathway and a downstream transcription factor SKN-1, an ortholog of mammalian Nrf, but not via DAF-16 (*Komura et al., 2013*). The PMK-1 pathway is also activated by *Lb. acidophilus* and *Lactobacillus fermentum* (*Kim et al., 2012*; *Park et al., 2018*). Animals fed a lactic acid bacterium, *Weissella*, show higher expression of *daf-16*, *aak-2*, and *jnk-1*, and extend lifespan in these gene-dependent manners (*Lee et al., 2015*). the *daf-16* pathway is also involved in thermotaxis (*Murakami et al., 2005*; *Kodama et al., 2006*) and salt-avoidance behaviors (*Tomioka et al., 2006*). In our condition, *daf-16* did not strongly affect thermotaxis at D1, while it was necessary for the maintenance of thermotaxis of *Lb. reuteri*-fed aged animals. Moreover, the time-specific knockdown of *daf-16* showed that DAF-16 had a specific role in the *Lb. reuteri*'s effects during aging. Since *daf-16* functions in neurons (*Figure 8C*) and has neuron-specific targets (*Kaletsky et al., 2016*), differential expressions of these genes with different diet might affect thermotaxis behavior. Indeed, our transcriptome analysis revealed that differentially expressed genes between animals fed *E. coli* or homofermentative LAB and those fed heterofermentative LAB were enriched with DAF-16 targets. Interestingly, our transcriptome data suggested that DAF-16 was activated in animals fed *E. coli* or homofermentative LAB although heterofermentative *Lb. reuteri*-fed aged animals showed *daf-16* dependency on thermotaxis. The regulation of DAF-16 in specific neurons might be opposite to that in the whole body. Single-cell transcriptome analysis for aged animals fed different diet will address this question in the future (*Cao et al., 2017*).

As a possible signal from the intestine to neurons, we propose an involvement of the neuropeptides. For example, in the young animals, INS-11 neuropeptide functions as a signal from the intestine to neurons to regulates avoidance behavior (*Lee and Mylonakis, 2017*). In our case, the RNAseq data of the animals with different age and diet led us to analyze the neuropeptide pathway. We found that *aex-5* encoding proprotein convertase is necessary to reduce the thermotaxis ability of *E. coli*-fed aged animals. *aex-5* was originally found as a mutant with defecation defects (*Thomas, 1990*) and functions in the intestine to regulate defecation. Since *aex-5* regulates a variety of neuropeptides (*Husson et al., 2006*), some of those neuropeptides may decrease the thermotaxis ability when fed with *E. coli*. *Lb. reuteri* might reduce these neuropeptides to maintain the thermotaxis of aged animals.

## Bacterial screen to address age-dependent phenotypes

Even with *C. elegans*, it is challenging to address age-dependent neuronal phenotypes because powerful forward genetic screens are not readily applicable to aged animals. Our study showed that bacterial screen could be useful for generating phenotypic diversity and addressing underlying molecular mechanisms in aged animals. The bacterial screen has been applied to various *C. elegans* phenotypes. Watson et al. carried out unbiased mutant screens of *E. coli* and *C. aquatica* to identify bacterial genes that affect the 'dietary sensor' in *C. elegans*, which increases the GFP intensity when fed *Comamonas*; they found that mutations in genes involved in vitamin B12 biosynthesis/import increase *C. elegans* dietary sensor activity (*Watson et al., 2014*). Zhou et al. screened 13 LAB and found that *Lactobacillus zeae* protects *C. elegans* from enterotoxigenic *E. coli* (*Zhou et al., 2014*). Given that *C. elegans* has its natural microbiota (*Berg et al., 2016*; *Dirksen et al., 2016*; *Samuel et al., 2016*; *Zhang et al., 2017*), the nervous system of animals in a natural environment may be affected by complex bacteria. Indeed, a recent study has revealed that tyramine produced from commensal bacteria affects *C. elegans* avoidance behavior (*O'Donnell et al., 2020*). Hence, bacterial screens will provide a unique angle of understanding for *C. elegans* research.

## Materials and methods
### Worm maintenance and strains

*C. elegans* strains were maintained at 23°C on NGM plates with *E. coli*, OP50, as previously reported (*Brenner, 1974*), except CB1370, which was maintained at 15°C until they become L4. N2 (Bristol) was used as the wild type. The following mutant strains were used for thermotaxis assays: CB1370 *daf-2(e1370ts) III*, DA1116 *eat-2(ad1116) II*, CF1038 *daf-16(mu86) I*, IK0656 *tax-6(db60) IV*, NUJ69 *kmo-1(tm4529) V*, NUJ71 *nkat-1(ok566) X*, NUJ559 *daf-16(hq389[daf-16::gfp::degron]); ieSi57[eft-3p::TIR1::mRuby::unc-54 3'UTR+Cbr-unc-119(+)] II*, VC48 *kpc-1(gk8) I*, VC671 *egl-3(ok979) V*, CB937 *bli-4(e937) I*, JT23 *aex-5(sa23) I*. NUJ69 *kmo-1(tm4529)* is a one-time outcrossed FX04529

*kmo-1(tm4529)* strain. NUJ71 *nkat-1(ok566)* is a two-time outcrossed RB784 *nkat-1(ok566)* strain. The transgenic strains were generated as described below.

## CRISPR knockout

To generate a *daf-16* allele that affects only the exon of the b isoform, we used the Co-CRISPR strategy (**Kim et al., 2014**). tracrRNA, *daf-16* crRNA (5'-GUCAUGCCAGAUGAAGAACA-3'), and *dpy-10* crRNA (5'-GCUACCAUAGGCACCACGAG-3') were annealed and injected into N2 animals with *eft-3*p::Cas9::NLS-3'UTR(*tbb-2*) plasmids. After picking dumpy and/or roller F$_1$ animals on individual plates, the mutation of the *daf-16* locus was detected by PCR and confirmed using Sanger sequencing. We obtained *knj36* which introduced 7 bp deletion in the first exon of the *daf-16* b isoform, corresponding to the intron of other isoforms. NUJ298 *daf-16(knj36) I* was used for thermotaxis assays.

## Plasmids and single-copy insertion of transgenes

Single-copy insertions of transgenes were generated using Cas9-based homologous recombination following the established protocol (**Andrusiak et al., 2019**). *daf-16*bp::*daf-16*b (gDNA) was cloned into the pCR8 vector (Thermo Fisher Scientific) vector using Gibson Assembly (New England Biolabs) (pKEN891). pKEN891 was recombined with the pCZGY2729 repair template vector using the LR reaction (Gateway Technology, Thermo Fisher Scientific) to generate the pKEN1016 repair template vector. pKEN1016 contains homology arms for *cxTi10882* site on chromosome IV and *rps-0*p::Hyg$^R$ in addition to the *daf-16*b promoter (4.9 kbp) and *daf-16*b gDNA. To swap promoters of pKEN1016, XhoI and AgeI sites were introduced before and after the *daf-16*b promoter, respectively (pKEN973). pKEN973 was digested with XhoI and AgeI, and the promoter was substituted to *myo-2*p (1.0 kbp, pKEN976), *myo-3*p (2.4 kbp, pKEN975), or *rgef-1*p (3.5 kbp, pKEN974) using Gibson Assembly.

We injected the repair template described above and *eft-3*p::Cas9+*cxTi10882* sgRNA (pCZGY2750) into N2 or CF1038 *daf-16(mu86)* with three red markers originally used for MosSCI insertion (**Frøkjaer-Jensen et al., 2008**): *rab-3*p::mCherry (pGH8), *myo-2*p::mCherry(pCFJ90), and *myo-3*p::mCherry (pCFJ104). The animals with the single-copy insertion were selected based on the hygromycin resistance (Hyg$^R$) and the absence of red fluorescence. The following strains were used for the rescue experiments of thermotaxis assays: NUJ306 *daf-16(mu86) I; knjSi17[daf-16bp::daf-16b] IV*, NUJ368 *daf-16(mu86) I; knjSi19[myo-2p::daf-16b] IV*, NUJ373 *daf-16(mu86) I; knjSi18[myo-3p::daf-16b] IV*, NUJ372 *daf-16(mu86) I; knjSi22[rgef-1p::daf-16b] IV*.

## Bacterial plates

*E. coli*, OP50 or HT115, was inoculated into Super Broth (32 g Bacto Tryptone (BD), 20 g Bacto Yeast extract (BD), 5 g NaCl (Wako), 5 mL 1 M NaOH in 1 L water) and cultured overnight at 37°C. LAB strains, provided by Megmilk Snow Brand company (**Supplementary file 2**), were inoculated into the liquid medium from glycerol stocks and cultured in the conditions described in **Supplementary file 2**. Bacterial cells were collected by centrifugation at 7000 × *g* for 10 min at 4°C. Cells were washed twice with sterile 0.9% NaCl solution. The washed bacteria were adjusted to a final concentration of 0.1 g/ml (wet weight) in NG buffer (25 mM K-PO$_4$ (pH 6), 50 mM NaCl, 1 mM CaCl$_2$, 1 mM MgSO$_4$). For heat killing, 0.1 g/ml bacteria in tubes were incubated for 1 hr in a 65°C incubator or 10 min in boiled water. By this treatment, bacterial colony-forming unit (cfu) became <1.0 × 10$^2$ cfu/ml, which is at least 10$^8$ lower than live bacteria (>1.5 × 10$^{10}$). For the mixed condition, bacteria were mixed to make the final concentration of 0.1 g/ml in total before spread on NGM plates. To crush bacteria, bacterial suspension was vigorously vibrated with glass beads at 4200 rpm for 15 cycles of 30 s ON and 30 s OFF using a bead-based homogenizer (MS-100R, Tomy). Two hundred microliters of the bacterial suspension were spread onto 60 mm NGM plates and dried overnight. NGM plates with peptone were used except for thermotaxis to see the effect of peptone and lifespan assays, in which NGM plates without peptone were used.

## Preparation of aged animals fed different bacteria

For behavioral assays, synchronized eggs were prepared by bleaching gravid hermaphrodites using 0.5× household bleach in 0.5 M NaOH and placed onto NGM plates with OP50. The eggs were cultivated at 23°C for 72 hr to obtain day one adults (D1). For the thermotaxis of *daf-2* mutants, eggs of N2 and CB1370 *daf-2(e1370)* were incubated at 15°C for 96 hr and subsequently at 23°C for 24 hr

to obtain D1. D1 animals were washed with NG buffer and transferred to NGM plates with OP50 or LAB every day for thermotaxis of aged animals. To expose animals to the bacterial odor, NG buffer as control or 0.1 g/ml bacterial solution was spotted on the lid of bacterial plates described above. Animals were cultivated on the upside-down plates as described in *Figure 5D*. For the thrashing and locomotion assay, animals were transferred individually by picking instead of washing. For the AID experiments, 4 mM auxin was added to the NGM plates as previously reported (*Zhang et al., 2015*), and animals were treated with auxin from eggs to D1 and/or D1 to D5.

## Thermotaxis assay

Animals were cultivated at 23°C ($T_{cult}$ = 23°C) unless otherwise noted. Population thermotaxis assays with a linear thermal gradient were performed as described (*Ito et al., 2006*). Fifty to 250 animals on cultivation plates were washed with M9 and placed at the center of the assay plates without food and with a temperature gradient of 17–23 or 20–26°C. The temperature gradient was measured to be ~0.5°C/cm. After letting them move for 1 hr, the animals were killed by chloroform. The number of adult animals in each of eight sections along the temperature gradient (*Figure 1A*) was scored under a stereomicroscope. The fraction of animals in each section was plotted on histograms. The performance and thermotaxis indices were calculated, as shown in *Figures 1A and 3C*, respectively. For temperature shift assays, $T_{cult}$ was shifted 24 h prior to the assay.

## Lifespan assay

Animals were synchronized by bleaching gravid adults and grown with regular NGM plates with OP50 until D1. D1 animals were washed three times with M9 buffer and transferred to peptone-free NGM plates supplemented with 50 mg/ml OP50 or LAB. Animals were transferred to new plates every day until they became D4 and every other day afterward. Dead animals were defined as having no voluntary movement after several touches on the head and tail and counted every day. Four independent sessions with 25 animals per session were combined for each condition.

## AFD and AIY imaging

To examine if AFD and AIY experience cell death at D5, NUJ296 *knjls15[gcy-8Mp::GCaMP6m+gcy-8Mp::tagRFP+ges-1p::tagRFP]* and IK1144 *njls26[AIYp::GCaMP3+AIYp::tagRFP+ges-1p::tagRFP]* were imaged, respectively. D1 and D5 animals were immobilized using 1 mM levamisole in M9, mounted on an agarose pad, and imaged using an Axio Imager.A2 equipped with a Plan-Apochromat ×63/1.4 oil objective (Zeiss). TagRFP signals in the cell body of AFD or AIY were visualized by green LED (555/30 nm) of Colibri 7 light source, and a quad-band path filter ser 90 HE LED (Zeiss) and used to evaluate possible cell death. We did not observe any loss of cell bodies in young or aged animals.

## Food recognition assay

The food recognition assay was performed as previously described with a few modifications (*Sawin et al., 2000*). Assay plates with food were prepared by spreading OP50 onto NGM plates. For well-fed animals, animals were washed twice in S basal buffer (*Brenner, 1974*) and transferred using a capillary glass pipette into a drop of the buffer on an assay plate with or without food. Five minutes after transfer, the number of body bends in 20-s intervals was counted. For starved animals, 5–15 animals were washed twice in S basal buffer and incubated on NGM plates without food for 30 min. After transferring them on assay plates with or without food, we measured the number of body bends.

## Salt-avoidance assay

For a gradient assay of salt chemotaxis (*Saeki et al., 2001*), a salt gradient was formed overnight by placing an agar plug containing 100 mM of NaCl (5 mm diameter) 2 cm away from the edge of the 90 mm assay plate (2% Bacto Agar, 5 mM K-PO$_4$ [pH 6.0], 1 mM CaCl$_2$, 1 mM MgSO$_4$). D1, *E. coli*-fed D5, and *Lb. reuteri*-fed D5 animals were divided into three groups. The first group of animals received no conditioning (Naive). For conditioning, animals were washed three times with chemotaxis buffer (5 mM KPO$_4$ [pH 6.0], 1 mM CaCl$_2$, 1 mM MgSO$_4$), transferred to the same buffer with 20 mM NaCl (NaCl-conditioned) or without NaCl (Mock-conditioned), and incubated at 25°C for 1 hr. These animals were placed at the center of the assay plates and then incubated at 25°C for 30 min. The

chemotaxis index was calculated as $(N_{NaCl} - N_{control})/(N_{total} - N_{origin})$ as indicated in *Figure 1—figure supplement 4B*. One hundred to two hundred animals were used in each assay.

For a quadrant assay of salt taxis (*Wicks et al., 2000*), we used a compartmentalized plate (Falcon X, Becton Dickinson Labware) as an assay plate (2% Bacto Agar, 5 mM K-PO$_4$ [pH 6.0], 1 mM CaCl$_2$, 1 mM MgSO$_4$). The plates were freshly prepared on the day of the assay with two different agar solutions, 0 and 25 mM NaCl. D1, *E. coli*-fed D5, and *Lb. reuteri*-fed D5 animals were divided into three groups. The first group of animals received no conditioning (Naive). For conditioning, animals were washed three times with chemotaxis buffer (5 mM K-PO$_4$ [pH 6.0], 1 mM CaCl$_2$, 1 mM MgSO$_4$), transferred to the same buffer with 100 mM NaCl (NaCl-conditioned) or without NaCl (Mock-conditioned), and incubated at room temperature for 15 min. These animals were placed at the center of the assay plates and then incubated at room temperature for 10 min. The chemotaxis index was calculated as $(N_{NaCl} - N_{control})/(N_{total} - N_{origin})$ as indicated in *Figure 1—figure supplement 4B*. One hundred to two hundred animals were used in each assay.

## Thrashing assay

A thrashing assay was performed, as previously described with a few modifications (*Tsalik and Hobert, 2003*). Animals were washed with NG buffer and transferred with a drop of NG buffer onto an NGM plate without food using a capillary glass pipet. In liquid, animals show lateral swimming movements (thrashes). We defined a single thrash as a complete movement through the midpoint and back and counted the number of thrashes for 30 s.

## Motility assay

Assay plates were prepared by placing circular filter paper with a one-inch hole on NGM plates with OP50 or LAB and soaking the paper with ~100 µl of 100 mM CuCl$_2$. A single animal was transferred to an assay plate with the cultured bacteria and left at 23°C for 3 min. The images of the bacterial lawn were captured by a digital camera (Fujifilm) through an eyepiece of a stereomicroscope, Stemi 508 (Zeiss). The trajectory of an animal on the lawn was traced using FIJI (*Schindelin et al., 2012*) and measured as the distance of locomotion.

## FITC labeling of bacteria

To examine whether *C. elegans* ingests bacteria, we used fluorescently labeled bacteria. Bacterial cells were incubated with phosphate-buffered saline (PBS, Takara) (unlabeled) or 0.1 mg/ml FITC-I (Wako) in PBS for 1 hr, washed with PBS three times, and resuspended with PBS at 0.1 g/ml. For the mixed condition, an equal amount of unlabeled and FITC-labeled bacteria were mixed. Bacteria were spread on NGM plates and dried. D1 adult animals were placed on the bacterial plate and incubated at 23°C for 20 min. Excess amounts of fluorescent bacteria were removed by letting worms crawl on an NGM plate without food for a few minutes. Animals were imaged after washing using an Axio Imager.A2 equipped with a Plan-Apochromat ×63/1.4 oil objective (Zeiss).

## Gram-staining

Bacteria are fixed with methanol and stained using Gram Color Kit (Muto Pure Chemicals Co, Ltd, Tokyo, Japan). Stained bacteria are imaged using an Axio Imager.A2 equipped with a Plan-Apochromat ×63/1.4 oil objective (Zeiss).

## Phylogenetic tree

16S rRNA sequences of LAB were obtained from the Genome database of NCBI (http://www.ncbi.nlm.nih.gov/genome/). The accession numbers are shown in *Supplementary file 2*. The phylogenetic tree was inferred by the Neighbor-Joining method based on the 16S rRNA gene sequence of model LAB strains. The evolutionary distances were computed using the Maximum Composite Likelihood method conducted in MEGA X (*Hall, 2013*).

## Quantitative RT-PCR

RNA was prepared as described in the RNA sequencing section. Two micrograms of total RNA were reverse transcribed to cDNA with a mixture of random and oligo dT primers using ReverTra Ace qPCR RT Master Mix with gDNA Remover (TOYOBO). The cDNA and gene-specific primers were used for

qPCR reaction with THUNDERBIRD SYBR qPCR Mix (TOYOBO), and the products were detected using a LightCycler 96 System (Roche). The gene-specific primers were used for *pha-4*: KN1370, 5′-GGTTGCCAGGTCCCCTGACA-3′; KN1371, 5′-GCCTACGGAGGTAGCATCCA-3′. *cdc-42* was used as a reference because it is stable and unaltered during aging (*Hoogewijs et al., 2008*; *Mann et al., 2016*) (KN1170, 5′-CTGCTGGACAGGAAGATTACG-3′; KN1171, 5′-CTCGGACATTCTCGAATGAA G-3′).

## RNA sequencing

Non-gravid young adult animals were used to avoid the effect of eggs inside the body. D5 animals fed *E. coli* or LAB (*Lb. gasseri*; *Lb. delbrueckii*, *P. pentosaceus*, *Lb. reuteri*, *Lb. rhamnosus*, and *Lb. plantarum*) were prepared as described above. Total RNA was extracted from whole animals using RNAiso Plus reagent (Takara) and sequenced using NovaSeq 6000 System by Macrogen Corp. Japan. We detected 13329 genes. The heatmap of the one-way hierarchical clustering was generated using *Z*-score for normalized value based on log2 by Macrogen. The fragments per kilobase of exon per million mapped fragments (FPKM) of D1, D5 (*E. coli*), and D5 (LAB, 6 in total) were used to perform the principal component analysis based on the variance–covariance matrix using R program. The eigenvalues of each transcript were ranked by the percentage of explained variance for each principal component, and the top and bottom 5% (666 genes each) were used to perform GO analysis using Metascape (*Zhou et al., 2019b*).

To further analyze the difference between D5 animals fed *E. coli* or homofermentative LAB (*Lb. gasseri*; *Lb. delbrueckii*) and D5 animals fed heterofermentative LAB (*P. pentosaceus*, *Lb. reuteri*, *Lb. rhamnosus*, and *Lb. plantarum*), we compared the expression of genes between these two groups and extracted differentially expressed genes as those with p value <0.05 by Student's *t*-test. Those differentially expressed genes were used to perform GO analysis using Metascape and compared with DAF-16-regulated genes (*Tepper et al., 2013*). Gene enrichment was calculated using Gene Set Enrichment Analysis (GSEA) (*Subramanian et al., 2005*).

## Statistical analyses

Box-and-whisker plots represent medians as center lines; boxes as first and third quartiles; whiskers as maximum and minimum values except for outliers, which are 1.5 times greater than the upper limit or 1.5 times smaller than the lower limit of the interquartile range; dots as outliers. In some figures, all the data points were overlayed on the box-and-whisker plots. We used Student's *t*-test to compare two samples and one- or two-way analysis of variance, followed by Dunnett's or Tukey–Kramer test to compare multiple samples using R (R core team, https://www.R-project.org/, Vienna, Austria) or GraphPad Prism 7.0 (GraphPad Software, La Jolla, CA). In all figures, *p < 0.05, **p < 0.01, ***p < 0.001. p > 0.05 is considered as not significant (ns).

## Acknowledgements

Megmilk Snow Brand Company supported this work. We thank members of the Nutritional Neuroscience laboratory and the Mori laboratory for their comments on the manuscript. Wei Huang and Pauline Rouillard helped with basic experiments. *C. elegans* mutant strains were provided by *Caenorhabditis* Genetics Center (CGC), funded by the NIH Office of Research Infrastructure Programs (P40 OD010440), and Dr. Shoehei Mitani of the National Bioresource Project of Japan. Dr. Yishi Jin provided pCZGY2729 and pCZGY2750 plasmids.

## Additional information

### Competing interests

Satoshi Higurashi: SH is a former employee of MEGMILK SNOW BRAND Co., Ltd. Sachio Tsukada: ST is a former employee of MEGMILK SNOW BRAND Co., Ltd. Masaru Tanaka: MT is an employee of MEGMILK SNOW BRAND Co., Ltd. The other authors declare that no competing interests exist.

## Funding

| Funder | Grant reference number | Author |
|---|---|---|
| Megmilk Snow Brand Co. Ltd. | | Kentaro Noma |

The funders had no role in study design, data collection, and interpretation, or the decision to submit the work for publication.

## Author contributions

Satoshi Higurashi, Data curation, Formal analysis, Investigation, Visualization, Methodology, Project administration; Sachio Tsukada, Data curation, Formal analysis, Investigation, Methodology, Project administration; Binta Maria Aleogho, Data curation, Formal analysis, Investigation, Visualization, Methodology; Joo Hyun Park, Formal analysis, Investigation, Methodology; Yana Al-Hebri, Investigation; Masaru Tanaka, Data curation, Formal analysis, Investigation, Visualization; Shunji Nakano, Conceptualization, Supervision, Project administration, Writing - review and editing; Ikue Mori, Conceptualization, Resources, Software, Supervision, Funding acquisition, Project administration, Writing - review and editing; Kentaro Noma, Conceptualization, Resources, Formal analysis, Supervision, Funding acquisition, Visualization, Methodology, Writing - original draft, Project administration, Writing - review and editing

## Author ORCIDs

Kentaro Noma ⓘ http://orcid.org/0000-0002-6487-8037

## Decision letter and Author response

Decision letter https://doi.org/10.7554/eLife.81418.sa1
Author response https://doi.org/10.7554/eLife.81418.sa2

# Additional files

## Supplementary files

• Supplementary file 1. Lifespan statistics. (a) Lifespan statistics for *Figure 1—figure supplement 1*. (b) Lifespan statistics for *Figure 4A*. (c) Lifespan statistics for *Figure 7C*.

• Supplementary file 2. List of LAB strains.

• Supplementary file 3. RNAseq analyses of aged animals fed with different bacteria. (a) RNAseq data. (b) Top 5% genes positively contributing to PC1. (c) Top 5% genes negaitively contributing to PC1. (d) Gene ontology analysis of top 5% genes positively contributing to PC1. (e) Gene ontology analysis of top 5% genes negatively contributing to PC1. (f) Top 5% genes positively contributing to PC2. (g) Top 5% genes negatively contributing to PC2. (h) Gene ontology analysis of top 5% genes positively contributing to PC2. (i) Gene ontology analysis of top 5% genes negatively contributing to PC2. (j) Genes enriched in *E. coli* or Homo lactic acid bacteria (LAB)-fed D5. (k) Gene ontology analysis of *E. coli* or Homo LAB-enriched genes. (l) Genes enriched in Hetero LAB-fed D5. (m) Gene ontology analysis of Hetero LAB-enriched genes.

• Transparent reporting form

## Data availability

RNA sequencing data has been deposited at NCBI (PRJNA968058).Other numerical data is available as Source data in the Excel format.

The following dataset was generated:

| Author(s) | Year | Dataset title | Dataset URL | Database and Identifier |
|---|---|---|---|---|
| Noma K | 2023 | Lactic acid bacteria-fed aged *C. elegans* | https://www.ncbi.nlm.nih.gov/bioproject/PRJNA968058 | NCBI BioProject, PRJNA968058 |

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
