## [Editor Report]

This important work focuses on the impact of diet on age-dependent behavior decline, showing that worms grown on *E. coli*, a common laboratory diet, lose their thermotaxis ability as they grow older, and that a diet of LAB partially rescue this effect. The evidence supporting the claims is solid, although the mechanism for the effects is not yet fully characterized. The work will be of interest to scientists interested in aging, behavior, diet, and potentially the microbiome.

---

## [Decision Letter]

**Decision letter after peer review:**

[Editors’ note: the authors submitted for reconsideration following the decision after peer review. What follows is the decision letter after the first round of review.]

Thank you for submitting your work entitled "*Lactobacilli* in a clade ameliorate age-dependent decline of thermotaxis behavior in *Caenorhabditis elegans*" for consideration by *eLife*. Your article has been reviewed by 2 peer reviewers, one of whom is a member of our Board of Reviewing Editors, and the evaluation has been overseen by a Senior Editor. The reviewers have opted to remain anonymous.

Our decision has been reached after consultation between the reviewers. Based on these discussions and the individual reviews below, we regret to inform you that your work will not be considered further for publication in *eLife*.

Summary:

This manuscript examines how lactic acid producing *E. coli* impact age-related decline in neurological function through the use of temperature-food associative learning or thermotaxis. In particular, the authors screen a panel of different lactate producing *E. coli* and identify a particular clade of bacteria, *Lactobacilli*, that are able to suppress age-dependent decline in thermotaxis in a daf-16 dependent manner. Moreover, they uncouple improvement in neurological function from lifespan determination and locomotion. Overall, this manuscript represents an interesting phenomenon regarding the effects of the lactic acid producing bacteria. However, it is not clear what is happening in the worm to elicit this neurological response and much work remains to determine this mechanism of action.

The reviewers each appreciate the careful nature of these worm behavioral assays including a host of different controls. It is interesting that a clade of lactic acid bacteria (LAB) can improve associative learning in *C. elegans*, and that many LAB strains of the same clade can improve thermotaxis in older nematodes, despite disparate results on longevity. However, there were some questions remaining about methodology, and more importantly, there is very little evidence provided on what the molecular mechanism might be behind this phenomenon. The final figure of the manuscript was quite limited, as it only briefly touches on molecular mechanism (only to give DAF-16 dependence). Since it has previously been shown that daf-16 mutant animals impact taste avoidance learning (Nagashima et al. PLOS Genetics, 2019, which is not cited), the dependence of DAF-16 and its role in associative learning seemed predictable. Overall, this study contains interesting observations that are not thoroughly enough developed for publication in *eLife*.

1) Data regarding dietary restriction and the eat-2 mutation appear to be misinterpreted. Thus, more attention and analysis should be dedicated to the effects of dietary restriction on their paradigm. It was interesting that a clade of LAB consistently reduced expression of PHA-4 transcription factor and the authors might benefit for expanding upon this observation. This is another avenue that could in principal lead toward a better mechanistic understanding.

2) In addition to molecular characterization, the manuscript provides little explanation at the cellular level. It is unclear what neurons or neuronal circuit are responsible for this phenomenon. Although mentioned in the discussion, this manuscript would benefit by close examination of the thermosensory circuit including the AFD and AIY neurons. How are these lactic acid producing *E. coli* ultimately signaling to the neurons? Do the LAB slow the rate of degeneration of either neuron? Is this phenomenon the result of lactic acid production or something else in the bacteria? Would it be possible to supplement lactic acid to worm media and produce the same result?

3. How is LAB different from Ecoli? Does metabolic composition of LAB dictate its impact on thermotaxis behavior of worms? In the manuscript the authors argue that LAB are a "better" food source than *E. coli*. How does one define better for something as broad as a food source? There is a difference here but it is very unclear what aspects of LAB physiology may play a role.

4. Does this phenomenon require eating LAB, or just perceiving it? The assays did not test whether perception of LAB diet is sufficient for its effect on thermotaxis, rather whether more time on LAB leads to better thermotaxis.

5. Showing a potential daf-16 interaction is plausible, given that daf-16 interacts with many key pathways in the worm and from the above referenced publications, but it is unclear whether this interaction is direct or indirect, or whether daf-16 is a major player in this pathway or just necessary for maintenance of health. What sensory pathways are activated when worms are fed on LAB diet, and how it finally interacts with daf-16?

*Reviewer #1:*

These investigators examine how lactic acid producing *E. coli* impact age-related decline in neurological function through the use of temperature-food associative learning or thermotaxis. In particular, they screen a panel of different lactate producing *E. coli* and identify a particular clade of bacteria, *Lactobacilli*, that are able to suppress age-dependent decline in thermotaxis in a daf-16 dependent manner. Moreover, they uncouple improvement in neurological function from lifespan determination and locomotion. Overall, this group presents an interesting phenomenon regarding the effects of the lactic acid producing bacteria. However, it is not clear what is happening in the worm to elicit this neurological response and much work remains to determine this mechanism of action.

While I can appreciate the careful nature of these worm behavioral assays including a host of different controls, these studies lack cellular and molecular details, which reduce my overall excitement for the story. It is interesting that a clade of lactic acid bacteria (LAB) can improve associative learning in *C. elegans*. However, I was very underwhelmed when I got to the final figure, which very briefly touched on molecular mechanism (only to give DAF-16 dependence). Since it has previously been shown that daf-16 mutant animals impact taste avoidance learning (Nagashima et al. PLOS Genetics, 2019), the dependence of DAF-16 and its role in associative learning seemed predictable. For future submissions, this previous study on DAF-16 should be referenced in the manuscript. Moreover, data regarding dietary restriction and the eat-2 mutation appear to be misinterpreted. Thus, more attention and analysis should be dedicated to the effects of dietary restriction on their paradigm. I thought that it was interesting that a clade of LAB consistently reduced expression of PHA-4 transcription factor and the authors might benefit for expanding upon this observation.

In addition to molecular characterization, the manuscript provides little explanation at the cellular level. It is unclear what neurons or neuronal circuit are responsible for this phenomenon. Although mentioned in the discussion, this manuscript would benefit by close examination of the thermosensory circuit including the AFD and AIY neurons. How are these lactic acid producing *E. coli* ultimately signaling to the neurons? Do the LAB slow the rate of degeneration of either neuron? Is this phenomenon the result of lactic acid production or something else in the bacteria? Would it be possible to supplement lactic acid to worm media and produce the same result?

At present, I believe that this manuscript is not acceptable for publication in *ELife*. However, this is an interesting phenomenon and more in-depth cellular and molecular characterization would warrant consideration for publication.

*Reviewer #2:*

This manuscript, "*Lactobacilli* in a clade ameliorate age-dependent decline of thermotaxis behavior in *Caenorhabditis elegans*," is focused on the impact of diet on age-dependent behavioral decline. The authors have utilize a thermotaxis screen using different lactic acid bacteria (LAB) and identify strains of LAB with the ability to ameliorate age dependent decline in thermotaxis behavior. The study introduces some interesting results, including the finding that many LAB strains of the same clade can improve thermotaxis in older nematodes, despite disparate results on longevity. However, there were some questions remaining about methodology, and more importantly, there is very little evidence provided on what the molecular mechanism might be behind this phenomenon. Overall, this study contains interesting findings that are not thoroughly enough developed for publication in *eLife*.

1. How is LAB different from Ecoli? Does metabolic composition of LAB dictate its impact on thermotaxis behavior of worms? In the manuscript the authors argue that LAB are a "better" food source than *E. coli*. How does one define better for something as broad as a food source? There is a difference here but it is very unclear what aspects of LAB physiology may play a role.

2. Does this phenomenon require eating LAB, or just perceiving it? The assays did not test whether perception of LAB diet is sufficient for its effect on thermotaxis, rather whether more time on LAB leads to better thermotaxis.

3. Showing a potential daf-16 interaction is plausible, given that daf-16 interacts with many key pathways in the worm, but it is unclear whether this interaction is direct or indirect, or whether daf-16 is a major player in this pathway or just necessary for maintenance of health. What sensory pathways are activated when worms are fed on LAB diet, and how it finally interacts with daf-16?

4. Similarly, the pha-4 and eat-2 data are interesting, but are not developed in any way. This is another avenue that could in principal lead toward a better mechanistic understanding.

[Editors’ note: further revisions were suggested prior to acceptance, as described below.]

Thank you for resubmitting your work entitled "Diets affect the age-dependent decline of associative learning in *Caenorhabditis elegans*" for further consideration by *eLife*. Your revised article has been evaluated by Timothy Behrens (Senior Editor) and a Reviewing Editor.

The manuscript has been improved but there are some remaining issues that need to be addressed, as outlined below:

Essential revisions:

Summary: The reviewers agreed that the data were strengthened in this revised version, but that the mechanistic insights remain lacking, mainly because the daf-16 data are unclear and there were concerns that daf-16 may be necessary but not a primary mechanism. Therefore, in addition to providing the missing figure panel and tables for all of their transcript lists, the most essential revision is to obtain something cleaner on the mechanism, either in the worms by making daf-16 more convincing/clear or identifying and validating another pathway through RNA-seq or from the bacteria (identifying key metabolites). As described in the reviews, both reviewers liked the assay and thought the phenomenon was much better described, leaving only some clear mechanistic insight remaining. Please note that the entire mechanism is not expected, but that the current hypothesis of daf-16 needs to be reworked or replaced

1) Rework/refine/replace mechanistic hypothesis, as the current data do not solidly support the daf-16 mechanism presented.

2) Provide missing figure and tables as described in the reviews.

*Reviewer #1 (Recommendations for the authors):*

The revised manuscript entitled "Diets affect the age-dependent decline of associative learning in *Caenorhabditis elegans*" is focused on the impact of diet on age dependent behavior decline. In the current study, worms grown on *E. coli*, a common laboratory diet, were found to lose their thermotaxis ability as they grew older. Moreover, they identified that heterofermentive LAB bacteria help the worms maintain a high thermotaxis capability, with varying results on lifespan. The revised manuscript did a good job addressing many of the finer points brought up in the previous review, some of which are listed below, but failed to provide solid mechanistic evidence as to either what about the LAB bacteria causes these changes or what in the worm is responsible for the effects. While it is not necessary to have both of these mechanisms nailed down, I found the support for the daf-16 mechanistic data was still weak and am still not confident that what role, if any, daf-16 plays in this age- and diet-related degradation of thermotaxis. While this is an interesting study with well-described phenomenology and the revised manuscript addresses many of the minor issues raised, the mechanistic data remain the primary weakness of the manuscript, and a stronger or more detailed understanding of the mechanism involved in LAB-mediated thermotaxis is necessary.

Things which are improved in the revised manuscript

1. The revised version addressed the perception issue and demonstrated that it did not play a role in LAB-mediated thermotaxis.

2. Necessity of neuronal daf-16 is new in the revised manuscript

3. Heat killed LAB bacteria still helps in the thermotaxis ability, suggesting the possible role of metabolites.

There is still a major concern in the manuscript about the mechanism. As mentioned in the previous review, there are at least two clear approaches toward mechanism (what is important about the bacteria, or what is important in the worm). For the bacteria, the authors nicely show that smell is not likely to be involved and that dead LAB bacteria can replicate the effect, but go no further. In the worm, the authors chose to look at genes which are involved in worms, and again honed in on daf-16. Unfortunately, while the daf-16 KO data look promising at first, much of the rest of the data make me question whether daf-16 actually plays a direct role in this phenotype in WT worms, or whether it is just necessary to have daf-16 in developing neurons to properly develop key neurons like the AIY, as has been previously published. The evidence from daf-2 animals and from RNA seq data both point toward a repressive role of daf-16 for thermotaxis, which is consistent with daf-16 only being important during development. In short, the current manuscript does not show daf-16 activation or requirement during adulthood, or daf-16 targets that are required, leaving the possibility of developmental or other secondary effects being key to daf-16's necessity, while other mechanisms cause the interesting phenotype observed.

1. In contrast to LAB bacteria, *E. coli* bacteria that were killed showed better thermotaxis performance with age. It is therefore possible that thermotaxis behavior is negatively influenced by pathogenic response to *E. coli* that is absent in LAB bacteria?

2. The revised manuscript showed the necessity of neuronal daf-16 in regulating high thermotaxis ability of LAB fed aged animals. But it is still not clear whether/how LAB bacteria might activate the neuronal daf-16.

3. The transcriptomics signatures clusters the LAB bacteria together irrespective of the fermentative mode of LAB, suggesting that transcriptomic signaling may not be important for the thermotaxis behavior.

4. As mentioned above, the transcriptional changes in the worms grown on heterofermentative LAB bacteria do not depend on daf-16, since the directionality of the transcription changes is in the same direction as the daf-16 mutant. It contradicts the role of daf-16-mediated signaling for LAB-mediated higher thermotaxis behavior. Similarly, daf-2 mutant worms grown on *E. coli* and LAB bacteria don't show significant differences in thermotaxis, suggesting the possibility that activating daf-16 may blunt this effect. This brings the possibility that daf-16 negatively regulates thermotaxis in aged animals and that its requirement could be an artifact of daf-16's role in neural development.

To summarize, the authors should either identify something about the bacteria (e.g., metabolites, pathogenicity or lack thereof) or solidify/address signaling pathways that involve neuronal Daf-16 or another mechanism when fed LAB bacteria.

Other points:

1. Figure 8C is missing/blank in the submission.

2. Lines 256-258 describe data about daf-16b that are not listed or in the figures from what I could find.

*Reviewer #2 (Recommendations for the authors):*

In this study, Higurashi et al. investigated how diet affects age-dependent thermotaxis behavioral decline in *C. elegans* and the underlying mechanism involved. Using various behavioral studies, they showed that changing the animal's diet during aging, from *E. coli* to lactic acid bacteria (LAB), allowed the animals to retain a high thermotaxis performance index comparable to control. The authors demonstrated that this increase in performance index when animals are switched to a LAB diet is independent of thermophilicity, strength of association of lactic acid bacteria during learning, lifespan, motility, or starvation. By contrast, factors such as age, diet, and nutritional value of diet all seem to have an observable effect on age-dependent thermotaxis behavioral decline. Of the 35 lactic acid bacteria investigated within this study, the authors demonstrated that daf-16, which functions in the neurons, was necessary for the maintenance of thermotaxis of animals that were fed a Lb. reuteri diet during aging.

Overall, the authors were able to achieve their proposed aims and the data shown supports their main conclusions.

A strength of this study is the use of a vast array of behavioral assays (thermotaxis performance, motility, lifespan, nutrition, dietary restriction, chemotaxis) to assess the age-dependent decline in temperature-food associated learning in *C. elegans*. The use of a wide variety of behavioral assays to support this hypothesis is worthy of mention. Additionally, the figures are nicely constructed and easy for a reader in any field to understand. The methods have been described in sufficient detail and clarity to allow for ease of replication and use by the scientific community.

Lacking within this study is a more in-depth understanding of the molecular mechanism at play. The DAF-16 interpretation is confusing. First, with respect to tissue specificity, why was DAF-16 function in the intestine not examined (isoform independent) as this tissue, in addition to the nervous system, has been shown to be important for DAF-16 function in several lifespan extension paradigms. Second, why is DAF-16 required for the associative learning with age, yet the best performing heterofermentative LAB show a down-regulation of DAF-16. Moreover, activation of DAF-16 via daf-2 mutants shows very low learning. Perhaps reduction, but not loss, of DAF-16 activity is driving this improved age-associated learning.

Perhaps to obtain more clarity on the mechanism underlying their LAB learning phenomena, this group can examine more deeply their RNAseq datasets and focus on transcriptional changes distinct for their best performing heterofermentative Clade A. According to the GO term analysis, lipid homeostasis and nutrient catabolism would be a great place to start. Two kynerinine metabolism genes are examined (specific for tryptophan), but perhaps a quick examination a few more that were implicated by their RNAseq would be appropriate. Please provide a list or table of the 71 transcripts enriched in hetero over homo Lb.

Overall this provides a very interesting phenomena on age-associated learning and implicates heterofermentative LAB but their proposed molecular mechanism of action is not well stated or transparent to the reader.

Line 1: Please revise the title to "Diet affects…."

2. Line 51-52: Please revise to "Age-dependent memory decline in the food-butanone association is ameliorated in the mutant of nkat-1 (Vohra, 53 Lemieux, Lin, & Ashrafi, 2018)".

3. The way the introduction section is currently written, the mention of daf-16 seems more of an afterthought. No clear introduction was given as to why including daf-16 within this study is pertinent for understanding the mechanism behind how diet affects age decline in *C. elegans*. A stronger case should be made for daf-16 within this section to show its relevance within the scope of this study.

4. Line 125: Please revise to "…behaviors".

5. Line 128: Revise to "…diet affects".

6. Line 146: revise to "(Figure S7A)".

7. Line 231: revise to "(Figure S11B)".

8. Consider utilizing the RNASeq analysis (mentioned later in the manuscript) of D5 worms verses D1 worms (control) fed either *E. coli* or the LAB diet. The authors state that they tested mutants of 3 genes (nkat-1, kom-1, daf-16) and selected these genes based on previous studies, but for a more comprehensive analysis, RNASeq analysis should be used to determine if there are other genes that are highly/more regulated during aging when fed the LAB diet. A heat map would be a good visual of all the genes that are up or down regulated. A nice compliment would be to perform qPCR on the 3 genes that were selected for analysis. A transcription profile of each gene for D1, D5 *E. coli*, D5 Lb. reuteri conditions would help bolster the case for selection of these 3 genes. A translation profile could also be explored using western blotting, but only if time permits.

9. Line 240: (Figure 7A) Curious as to why so few animals were analyzed for the kmo-1 and nkat-1 mutants compared to the WT and daf-16 mutants. Please explain this discrepancy.

10. Line 263: Figure 8C is missing from the figure panel.

11. Line 284: revise to "and biological processes".

12. Line 320: revise to "…diet affects".

13. Line 322: revise to "…aging and diet".

14. Line 329: revise to "…site".

15. Line 1040: Revise to "…at different temperatures" in the title.

16. Line 1044: Please revise to "…at different ages" in the title.

17. Figure 1: The performance index that is included in this figure is a bit confusing. The index equation includes nj and N but listed below in the definition of terms is ni. ni is not within the performance index equation. Was this an error? Please revise for clarity.

18. Figure S1, Figure 4, Figure 7: The authors indicated that survival curves were conducted at N=4, 25 animals/experiment. However, typically in the worm field, life spans are conducted at N=3, 100 animals/experiment to bolster statistical significance and a statistical table is also included for the lifespans. Please revise.

19. Figure S3: Why were animals only cultivated with OP50 at D1? Why not cultivate on both OP50 and HT115 in parallel for better comparison of the shift at D5 on the 2 bacteria?

---

## [Author Response]

[Editors’ note: the authors resubmitted a revised version of the paper for consideration. What follows is the authors’ response to the first round of review.]

The reviewers each appreciate the careful nature of these worm behavioral assays including a host of different controls. It is interesting that a clade of lactic acid bacteria (LAB) can improve associative learning in *C. elegans*, and that many LAB strains of the same clade can improve thermotaxis in older nematodes, despite disparate results on longevity. However, there were some questions remaining about methodology, and more importantly, there is very little evidence provided on what the molecular mechanism might be behind this phenomenon. The final figure of the manuscript was quite limited, as it only briefly touches on molecular mechanism (only to give DAF-16 dependence). Since it has previously been shown that daf-16 mutant animals impact taste avoidance learning (Nagashima et al. PLOS Genetics, 2019, which is not cited), the dependence of DAF-16 and its role in associative learning seemed predictable. Overall, this study contains interesting observations that are not thoroughly enough developed for publication in eLife.1) Data regarding dietary restriction and the eat-2 mutation appear to be misinterpreted. Thus, more attention and analysis should be dedicated to the effects of dietary restriction on their paradigm. It was interesting that a clade of LAB consistently reduced expression of PHA-4 transcription factor and the authors might benefit for expanding upon this observation. This is another avenue that could in principal lead toward a better mechanistic understanding.

We apologize for the lack of statistical analysis between D1 and D5 and insufficient explanation of the data. The original data had a large variation, so we repeated the experiment and added the statistical analysis to compare all the conditions. We did not change the conclusion and argue that eat-2 mutants do not have higher thermotaxis ability in aged *E. coli*-fed animals. Repeated analysis showed that young eat-2 animals showed the thermotaxis defects. Therefore, we weaken the statement and move the data to Figure S11.

We appreciate the comments on pha-4. Indeed, it would be interesting to test effect of the pha-4 loss-of-function. However, pha-4 is important for the development and a good allele for aging study is unavailable. RNAi might be another option, but we have not successfully introduced RNAi to analyze the behavior of aged animals, which has a large variation. Thus, we regret to say that the analysis of pha-4 is beyond the scope of this manuscript.

2) In addition to molecular characterization, the manuscript provides little explanation at the cellular level. It is unclear what neurons or neuronal circuit are responsible for this phenomenon. Although mentioned in the discussion, this manuscript would benefit by close examination of the thermosensory circuit including the AFD and AIY neurons.

By examining the tissue specificity, we revealed that daf-16 functions in neurons (Figure 8C). Therefore, diets clearly affected the nervous system. We conducted ca^2+^ imaging of AFD and AIY neurons but have not got clear conclusion about the effect on the neural circuit. Thus, we decided to focus on reporting the phenomena and the molecular mechanism in this manuscript.

How are these lactic acid producing *E. coli* ultimately signaling to the neurons?

Our additional experiments suggest that bacteria affect thermotaxis as nutrition, instead of affecting sensory neurons as smell (Figure 5D) Moreover, we carried out RNAseq analysis to examine the differentially expressed genes by age and diet and found that neuropeptide might be involved in the signaling from the intestine to neurons (Figures 9 and S12).

Do the LAB slow the rate of degeneration of either neuron?

Our additional experiment confirmed that AFD or AIY did not experience apoptosis in *E. coli* fed D5 animals (Figure S4). Moreover, Huang et al. reported the morphological defects of the AFD neurons did not correlate with the AFD function.

Is this phenomenon the result of lactic acid production or something else in the bacteria? Would it be possible to supplement lactic acid to worm media and produce the same result?

Lactic acid should not be sufficient because all of LAB are lactic acid producing and homo LAB has even higher lactic acid production but had no effect on the thermotaxis of aged worms. Therefore, we speculate that the bacteria-producing metabolites caused the phenotype.

3. How is LAB different from Ecoli?

We could not find a paper to compare the metabolites of *E. coli* and LAB in parallel. We carried out the analysis of possible difference in metabolites of those bacteria using genome information of type strains of bacteria and Rapid Annotation using Subsystem Technology (RAST). However, we did not include this analysis because we could not reach meaningful conclusion.

Does metabolic composition of LAB dictate its impact on thermotaxis behavior of worms?

We think so because the heat-killed Lb. reuteri had a similar effect to the live Lb. reuteri (Figure 5B). Since heterofermentative *Lactobacilli* and homofermentative *Lactobacilli* had clear difference of the effect on the thermotaxis of aged animals, difference of metabolic composition of those bacteria might be the key.

In the manuscript the authors argue that LAB are a "better" food source than *E. coli*. How does one define better for something as broad as a food source? There is a difference here but it is very unclear what aspects of LAB physiology may play a role.

We agreed with the reviewer and revised the statement not to argue that LAB is a “better”. Indeed, further experiment of the mixture of *E. coli* and Lb. reuteri revealed that *E. coli* had predominant effects (Figure 5C).

4. Does this phenomenon require eating LAB, or just perceiving it? The assays did not test whether perception of LAB diet is sufficient for its effect on thermotaxis, rather whether more time on LAB leads to better thermotaxis.

Our additional data support that eating is required. (1) We added an experiment to show that both *E. coli* and Lb. reuteri are ingested in animals using fluorescently labeled bacteria (Figure S9). (2) We added an experiment to show that perception of bacterial smell was not enough to recapitulate eating bacteria (Figure 5D). (3) Moreover, switching the bacteria on the last day on Figure 5A addressed it. *E. coli*-fed until D4 and LAB-fed on D5 did not increase the index suggesting that acute perception of LAB is not enough for better index.

5. Showing a potential daf-16 interaction is plausible, given that daf-16 interacts with many key pathways in the worm and from the above referenced publications, but it is unclear whether this interaction is direct or indirect, or whether daf-16 is a major player in this pathway or just necessary for maintenance of health.

We agree with the reviewer that we did not provide enough evidence to suggest how daf-16 works although daf-16 is expressed in many tissues and has various functions.

The effect of daf-16 is not simply due to the requirement for the general health because wild type and daf-16 showed the similar lifespan (Figure 7C) in the Lb. reuteri-fed condition.

Furthermore, we added an evidence suggesting that daf-16 functions in neurons (Figure 8C). However, daf-16’s effect is not thermotaxis per se given that daf-16 D1 can perform thermotaxis relatively well (Figure 7A). This evidence suggest that daf-16 plays a specific role in the nervous system rather than being necessary for the maintenance of general health.

What sensory pathways are activated when worms are fed on LAB diet, and how it finally interacts with daf-16?

We do not think sensory pathways are activated by LAB as indicated by a newly added smell experiment (Figure 5D). Rather, we think that ingestion of LAB change the internal state of animals during aging, which in turn signal to the nervous system. As we discussed above, we speculate that the gut-neuron interaction might be through neuropeptides.

[Editors’ note: what follows is the authors’ response to the second round of review.]

Essential revisions:Summary: The reviewers agreed that the data were strengthened in this revised version, but that the mechanistic insights remain lacking, mainly because the daf-16 data are unclear and there were concerns that daf-16 may be necessary but not a primary mechanism. Therefore, in addition to providing the missing figure panel and tables for all of their transcript lists, the most essential revision is to obtain something cleaner on the mechanism, either in the worms by making daf-16 more convincing/clear or identifying and validating another pathway through RNA-seq or from the bacteria (identifying key metabolites). As described in the reviews, both reviewers liked the assay and thought the phenomenon was much better described, leaving only some clear mechanistic insight remaining. Please note that the entire mechanism is not expected, but that the current hypothesis of daf-16 needs to be reworked or replaced1) Rework/refine/replace mechanistic hypothesis, as the current data do not solidly support the daf-16 mechanism presented.

We mainly added two points. One is refining the action of *daf-16* by providing evidence that *daf-16* functions during aging and not during development (Figure 8D). The other is that we focused on the neuropeptide signaling from the RNAseq data and propose it as a mechanism to link between the intestine and neurons by providing evidence of the involvement of a gene crucial for neuropeptide synthesis (Figure 9C).

2) Provide missing figure and tables as described in the reviews.

We added two figures, two supplentary figures, and 14 supplementary tables listed below:

Figure 8D Time-specific *daf-16* KD.Figure 9C Neuropeptide synthesis mutants.Figure 8—figure supplement 1 Time-specific *daf-16* KD control experiment.Figure 9—figure supplement 1 Clustering analysis of RNAseq data.Supplementary file 1 Lifespan statistics.Supplementary file 3a RNAseq data.Supplementary file 3b Top 5% genes positively contributing to PC1.Supplementary file 3c Top 5% genes negatively contributing to PC1.Supplementary file 3d Gene ontology analysis of top 5% genes positively contributing to PC1.Supplementary file 3e Gene ontology analysis of top 5% genes negatively contributing to PC1.Supplementary file 3f Top 5% genes positively contributing to PC2.Supplementary file 3g Top 5% genes negatively contributing to PC2.Supplementary file 3h Gene ontology analysis of top 5% genes positively contributing to PC2.Supplementary file 3i Gene ontology analysis of top 5% genes negatively contributing to PC2.Supplementary file 3j Genes enriched in *E. coli* or Homo LAB-fed D5.Supplementary file 3k Gene ontology analysis of *E. coli* or Homo LAB-enriched genes.Supplementary file 3l Genes enriched in Hetero LAB-fed D5.Supplementary file 3m Gene ontology analysis of Hetero LAB-enriched genes.

Further details for each point are provided below.

Reviewer #1 (Recommendations for the authors):The revised manuscript entitled "Diets affect the age-dependent decline of associative learning in Caenorhabditis elegans" is focused on the impact of diet on age dependent behavior decline. In the current study, worms grown on *E. coli*, a common laboratory diet, were found to lose their thermotaxis ability as they grew older. Moreover, they identified that heterofermentive LAB bacteria help the worms maintain a high thermotaxis capability, with varying results on lifespan. The revised manuscript did a good job addressing many of the finer points brought up in the previous review, some of which are listed below, but failed to provide solid mechanistic evidence as to either what about the LAB bacteria causes these changes or what in the worm is responsible for the effects. While it is not necessary to have both of these mechanisms nailed down, I found the support for the daf-16 mechanistic data was still weak and am still not confident that what role, if any, daf-16 plays in this age- and diet-related degradation of thermotaxis. While this is an interesting study with well-described phenomenology and the revised manuscript addresses many of the minor issues raised, the mechanistic data remain the primary weakness of the manuscript, and a stronger or more detailed understanding of the mechanism involved in LAB-mediated thermotaxis is necessary.Things which are improved in the revised manuscript1. The revised version addressed the perception issue and demonstrated that it did not play a role in LAB-mediated thermotaxis.2. Necessity of neuronal daf-16 is new in the revised manuscript3. Heat killed LAB bacteria still helps in the thermotaxis ability, suggesting the possible role of metabolites.There is still a major concern in the manuscript about the mechanism. As mentioned in the previous review, there are at least two clear approaches toward mechanism (what is important about the bacteria, or what is important in the worm). For the bacteria, the authors nicely show that smell is not likely to be involved and that dead LAB bacteria can replicate the effect, but go no further. In the worm, the authors chose to look at genes which are involved in worms, and again honed in on daf-16. Unfortunately, while the daf-16 KO data look promising at first, much of the rest of the data make me question whether daf-16 actually plays a direct role in this phenotype in WT worms, or whether it is just necessary to have daf-16 in developing neurons to properly develop key neurons like the AIY, as has been previously published. The evidence from daf-2 animals and from RNA seq data both point toward a repressive role of daf-16 for thermotaxis, which is consistent with daf-16 only being important during development. In short, the current manuscript does not show daf-16 activation or requirement during adulthood, or daf-16 targets that are required, leaving the possibility of developmental or other secondary effects being key to daf-16's necessity, while other mechanisms cause the interesting phenotype observed.1. In contrast to LAB bacteria, *E. coli* bacteria that were killed showed better thermotaxis performance with age. It is therefore possible that thermotaxis behavior is negatively influenced by pathogenic response to *E. coli* that is absent in LAB bacteria?

As the reviewer pointed out, *E. coli* is known to be mildly pathogenic to *C. elegans* during aging (Cabreiro and Gems, EMBO Mol. Med. (review), 2013). However, in our case, *E. coli* killed at 65C had the same effects as live *E. coli,* suggesting that the pathogenic effects of live *E. coli* are not the primary cause. To make this point clearer, we added the following sentence (underlined) in Line 373:

“In our thermotaxis assay on aged animals, *E. coli* and LAB killed by 65 °C treatment had similar effects to live bacteria. This result implies that, instead of the action of live bacteria, such as pathogenic effects of *E. coli* (Cabreiro & Gems, 2013), bacterial nutrition might be responsible for the effect on the thermotaxis of aged *C. elegans*.”

2. The revised manuscript showed the necessity of neuronal daf-16 in regulating high thermotaxis ability of LAB fed aged animals. But it is still not clear whether/how LAB bacteria might activate the neuronal daf-16.

To get an insight into how bacterial diet affects neurons, we focused on the neuropeptides enriched in our RNAseq analysis. To avoid redundancy of the effect of neuropeptides, we wanted to analyze the neuropeptide synthesis or secretion pathway. Although *unc-31* encoding CAPS involved in dense-core vesicle exocytosis is commonly used, we could not use it because of its locomotion defects. Therefore, we focused on the four proprotein convertases crucial for neuropeptide synthesis and found that *aex-5* known to function in the intestine is involved in decreasing the thermotaxis ability of *E. coli-*fed aged mutants. Although we cannot provide direct evidence to answer the reviewer’s question, this experiment suggests that neuropeptide signaling may be involved in the dietary effect on the thermotaxis as a link between the intestine and neurons. We added Figure 9C and texts in Results (Line 292) and Discussion (Line 436).

3. The transcriptomics signatures clusters the LAB bacteria together irrespective of the fermentative mode of LAB, suggesting that transcriptomic signaling may not be important for the thermotaxis behavior.4. As mentioned above, the transcriptional changes in the worms grown on heterofermentative LAB bacteria do not depend on daf-16, since the directionality of the transcription changes is in the same direction as the daf-16 mutant. It contradicts the role of daf-16-mediated signaling for LAB-mediated higher thermotaxis behavior. Similarly, daf-2 mutant worms grown on *E. coli* and LAB bacteria don't show significant differences in thermotaxis, suggesting the possibility that activating daf-16 may blunt this effect. This brings the possibility that daf-16 negatively regulates thermotaxis in aged animals and that its requirement could be an artifact of daf-16's role in neural development.

We agree with the reviewer that our transcriptome analysis did not capture the entire mechanism. This is mainly because both aging and different diet had pleiotropic effects on the entire body of the animals and changed the transcriptomic landscape of animals, including large tissues like the intestine. We showed that *daf-16* functions in the neurons in the thermotaxis of aged animals. Since neurons are relatively small, they had less impact on the transcriptomic data of the whole organism. Our data implied an interesting possibility that the *daf-16* regulates different genes between the neurons and other tissues. In the future, the transcriptome analysis of single-cell RNAseq can address this issue, but at this point, single-cell analysis is technically challenging to apply to adult animals and aged animals.

To summarize, the authors should either identify something about the bacteria (e.g., metabolites, pathogenicity or lack thereof) or solidify/address signaling pathways that involve neuronal Daf-16 or another mechanism when fed LAB bacteria.Other points:1. Figure 8C is missing/blank in the submission.

I’m sorry for our careless mistake. It was added.

2. Lines 256-258 describe data about daf-16b that are not listed or in the figures from what I could find.

It was because Figure 8C was missing in the previous version. It was added.

Reviewer #2 (Recommendations for the authors):In this study, Higurashi et al. investigated how diet affects age-dependent thermotaxis behavioral decline in *C. elegans* and the underlying mechanism involved. Using various behavioral studies, they showed that changing the animal's diet during aging, from *E. coli* to lactic acid bacteria (LAB), allowed the animals to retain a high thermotaxis performance index comparable to control. The authors demonstrated that this increase in performance index when animals are switched to a LAB diet is independent of thermophilicity, strength of association of lactic acid bacteria during learning, lifespan, motility, or starvation. By contrast, factors such as age, diet, and nutritional value of diet all seem to have an observable effect on age-dependent thermotaxis behavioral decline. Of the 35 lactic acid bacteria investigated within this study, the authors demonstrated that daf-16, which functions in the neurons, was necessary for the maintenance of thermotaxis of animals that were fed a Lb. reuteri diet during aging.Overall, the authors were able to achieve their proposed aims and the data shown supports their main conclusions.A strength of this study is the use of a vast array of behavioral assays (thermotaxis performance, motility, lifespan, nutrition, dietary restriction, chemotaxis) to assess the age-dependent decline in temperature-food associated learning in *C. elegans*. The use of a wide variety of behavioral assays to support this hypothesis is worthy of mention. Additionally, the figures are nicely constructed and easy for a reader in any field to understand. The methods have been described in sufficient detail and clarity to allow for ease of replication and use by the scientific community.Lacking within this study is a more in-depth understanding of the molecular mechanism at play. The DAF-16 interpretation is confusing. First, with respect to tissue specificity, why was DAF-16 function in the intestine not examined (isoform independent) as this tissue, in addition to the nervous system, has been shown to be important for DAF-16 function in several lifespan extension paradigms. Second, why is DAF-16 required for the associative learning with age, yet the best performing heterofermentative LAB show a down-regulation of DAF-16. Moreover, activation of DAF-16 via daf-2 mutants shows very low learning. Perhaps reduction, but not loss, of DAF-16 activity is driving this improved age-associated learning.

We understand that the intestine is the site of action for lifespan extension. However, we did not examine the rescue of the intestine because *daf-16b* that rescued the phenotype is not expressed in the intestine. We think that lifespan and behavioral aging have different mechanisms. The second point also makes sense. We speculate that the lack of tissue specificity caused this discrepancy. As discussed in the rebuttal for the first reviewer, the targets in the neurons important for thermotaxis may differ from those in the intestine, which is vital for the lifespan.

Perhaps to obtain more clarity on the mechanism underlying their LAB learning phenomena, this group can examine more deeply their RNAseq datasets and focus on transcriptional changes distinct for their best performing heterofermentative Clade A. According to the GO term analysis, lipid homeostasis and nutrient catabolism would be a great place to start. Two kynerinine metabolism genes are examined (specific for tryptophan), but perhaps a quick examination a few more that were implicated by their RNAseq would be appropriate. Please provide a list or table of the 71 transcripts enriched in hetero over homo Lb.

We added new Supplementary file 3j and 3l to list the genes found from the comparison between Hetero LAB vs. Homo LAB or *E. coli*. We also added supplementary tables regarding the analyses of RNAseq (Supplementary File 3a~3m). As the reviewer suggested, we have tested the effect of some conserved genes in the list whose mutants are available at the stock center (e.g., *comt-2(ok2244)*) and generated by ourselves using CRISPR (*arrd-6*). However, these mutants showed no phenotype in D1, *E. coli*-fed D5, or *Lb. reuteri*-fed D5. Instead, we focused on neuropeptide signaling and provided evidence that neuropeptide synthesis is involved, as discussed in the rebuttal for the first reviewer in detail.

Overall this provides a very interesting phenomena on age-associated learning and implicates heterofermentative LAB but their proposed molecular mechanism of action is not well stated or transparent to the reader.Line 1: Please revise the title to "Diet affects….".

Revised to “Bacterial diet affects….”.

2. Line 51-52: Please revise to "Age-dependent memory decline in the food-butanone association is ameliorated in the mutant of nkat-1 (Vohra, 53 Lemieux, Lin, & Ashrafi, 2018)".

Revised (removed the typo).

3. The way the introduction section is currently written, the mention of daf-16 seems more of an afterthought. No clear introduction was given as to why including daf-16 within this study is pertinent for understanding the mechanism behind how diet affects age decline in *C. elegans*. A stronger case should be made for daf-16 within this section to show its relevance within the scope of this study.

The following sentence was added to the introduction to show the implication of the *daf-16* involvement in the temperature-related behavior in *C. elegans* (Line 77).

“Moreover, *daf-16* is involved in the age-dependent modulation of isothermal tracking behavior in *C. elegans* with a regular *E. coli* diet (H. Murakami et al., 2005).”

4. Line 125: Please revise to "…behaviors".

Revised.

5. Line 128: Revise to "…diet affects".

Revised to “…bacterial diet affects”.

6. Line 146: revise to "(Figure S7A)".

Revised.

7. Line 231: revise to "(Figure S11B)".

Revised.

8. Consider utilizing the RNASeq analysis (mentioned later in the manuscript) of D5 worms verses D1 worms (control) fed either *E. coli* or the LAB diet. The authors state that they tested mutants of 3 genes (nkat-1, kom-1, daf-16) and selected these genes based on previous studies, but for a more comprehensive analysis, RNASeq analysis should be used to determine if there are other genes that are highly/more regulated during aging when fed the LAB diet. A heat map would be a good visual of all the genes that are up or down regulated. A nice compliment would be to perform qPCR on the 3 genes that were selected for analysis. A transcription profile of each gene for D1, D5 *E. coli*, D5 Lb. reuteri conditions would help bolster the case for selection of these 3 genes. A translation profile could also be explored using western blotting, but only if time permits.

The three genes were selected as the candidates based on the previous studies before performing RNAseq analysis. The expression of *daf-16* and *kmo-1* were changed less than 2 folds, and *nkat-1* was undetectable in RNAseq. To make it clear, we added the following sentence (Line 313):

“The expression of *daf-16* itself was not changed in the RNAseq (Supplementary file 3a).”

Moreover, based on the suggestion, we added the heatmap as Figure 8—figure supplement 1 and revised the text accordingly (Line 276).

9. Line 240: (Figure 7A) Curious as to why so few animals were analyzed for the kmo-1 and nkat-1 mutants compared to the WT and daf-16 mutants. Please explain this discrepancy.

We increased the sample size for *kmo-1* and *nkat-1*. The sample size of the wt is large because we have the same-day wt control for all the experiments for the mutants.

10. Line 263: Figure 8C is missing from the figure panel.

I’m sorry for our careless mistake. It is added.

11. Line 284: revise to "and biological processes".

Revised.

12. Line 320: revise to "…diet affects".

Revised.

13. Line 322: revise to "…aging and diet".

Revised.

14. Line 329: revise to "…site".

Revised.

15. Line 1040: Revise to "…at different temperatures" in the title.

Revised.

16. Line 1044: Please revise to "…at different ages" in the title.

Revised.

17. Figure 1: The performance index that is included in this figure is a bit confusing. The index equation includes nj and N but listed below in the definition of terms is ni. ni is not within the performance index equation. Was this an error? Please revise for clarity.

Yes, this was an error. Thank you for pointing it out. Revised.

18. Figure S1, Figure 4, Figure 7: The authors indicated that survival curves were conducted at N=4, 25 animals/experiment. However, typically in the worm field, life spans are conducted at N=3, 100 animals/experiment to bolster statistical significance and a statistical table is also included for the lifespans. Please revise.

Thank you for the advice. I understand that more sample size could be better. However, we obtained consistent data across four independent experiments for Figure 4 and Figure 1—figure supplement 1, and the data is clear (significance and n.s.). For Figure 7, we increased the sample size to be 150 animals. Moreover, examples of publications with ~100 animals can be found in the classic paper (Kenyon et al., Nature, 1993), technical paper (Sutphin and Kaeberlein, JoVE, 2009), and a recent paper (Frankino et al., JoVE, 2022). Therefore, we believe that our data is sufficient to support our conclusion. Based on the suggestion, we added the statistical table for the lifespan assay (Supplementary file 1).

19. Figure S3: Why were animals only cultivated with OP50 at D1? Why not cultivate on both OP50 and HT115 in parallel for better comparison of the shift at D5 on the 2 bacteria?

This is because we want to avoid the effect of bacteria on the development of animals. We aimed to examine the effect of bacteria during aging after growing them on the same bacteria, the regular diet OP50.